# Revisiting Block-based Quantisation:
# What is Important for Sub-8-bit LLM Inference?

**Cheng Zhang[1], Jianyi Cheng[1], Ilia Shumailov[2], George A. Constantinides[1], Yiren Zhao[1]**
[1]Imperial College London, [2]University of Oxford
{cheng.zhang122, jianyi.cheng17, g.constantinides, a.zhao}@imperial.ac.uk
ilia.shumailov@chch.ox.ac.uk

## Abstract

The inference of Large language models (LLMs) requires immense computation and memory resources. To curtail these costs, quantisation has emerged as a promising solution, but existing LLM quantisation mainly focuses on 8-bit. In this work, we explore the statistical and learning properties of the LLM layer and attribute the bottleneck of LLM quantisation to *numerical scaling offsets*. To address this, we adapt block quantisations for LLMs, a family of methods that share scaling factors across packed numbers. Block quantisations efficiently reduce the numerical scaling offsets solely from an arithmetic perspective, without additional treatments in the computational path. Our nearly-lossless quantised 6-bit LLMs achieve a $19\times$ higher arithmetic density and $5\times$ memory density than the `float32` baseline, surpassing the prior art 8-bit quantisation by $2.5\times$ in arithmetic density and $1.2\times$ in memory density, without requiring any data calibration or re-training. We also share our insights into sub-8-bit LLM quantisation, including the mismatch between activation and weight distributions, optimal fine-tuning strategies, and a lower quantisation granularity inherent in the statistical properties of LLMs. The latter two tricks enable nearly-lossless 4-bit LLMs on downstream tasks. Our code is open-sourced [1].

## 1 Introduction

Pre-trained Large Language Models (LLMs) (Brown et al., 2020; Black et al., 2021; Zhang et al., 2022) have demonstrated impressive performance on a range of Natural Language Processing (NLP) tasks. However, their underlying computational and memory costs are a critical bottleneck to their usability. For instance, the larger variants in the GPT family scale up to hundreds of billions of parameters, requiring at least 300GB of memory to store

these parameters in a float16 format (Brown et al., 2020). Quantisation serves as a natural solution for reducing the cost of running inference on these LLMs (Yao et al., 2022; Xiao et al., 2022; Dettmers et al., 2022), as a low-precision format enables cost savings across all relevant efficiency metrics: reduced on-chip memory, increased arithmetic intensity for matrix multiplies, and decreased DRAM bandwidth requirement. On the other hand, the growing popularity of running services such as ChatGPT (OpenAI, 2022) provides an impetus for exploring the use of custom silicon to support LLM inference. This raises the question: *What would a low-precision number system look like in these near-future LLM hardware accelerators (ASICs)?*

LLM quantisation is challenging because of the activations with large absolute magnitudes, also known as activation outliers (Bondarenko et al., 2021; Xiao et al., 2022). Previous approaches have proposed various techniques to address such outliers. However, these either require additional treatments in the integer quantisation domain (LLM.int8() and SmoothQuant) or yield unsatisfactory performance (ZeroQuant); and prior work has primarily focused on arithmetics that can be ported to GPUs. We observe that the presence of outliers necessitates different scaling factors at a finer granularity than per-tensor or per-token level (Yao et al., 2022; Xiao et al., 2022). This insight naturally leads us to revisit arithmetic systems with small exponents, such as MiniFloat (Sun et al., 2019), Block Minifloat (Fox et al., 2021), Block Logarithm (Miyashita et al., 2016), and Block Floating Point (Kalliojarvi and Astola, 1996), as they can effectively represent outliers in Transformer models. To the best of our knowledge, our work is the first to systemically investigate short-exponent arithmetics for LLM quantisation.

Figure 1 illustrates the variance of the tensors joining the GEMMs in an OPT-6.7B (Zhang et al.,

---

[1]https://github.com/ChengZhang-98/llm-mixed-q

| METHOD | (QW, QACT) | BITWIDTH | PTQ OR TAQ | # QUANTISED GEMMs |
|---|---|---|---|---|
| ZEROQUANT (YAO ET AL., 2022) | ($\sqrt{}$, $\sqrt{}$) | W4A8 | TAQ | 8/8 |
| LLM.INT8() (DETTMERS ET AL., 2022) | ($\sqrt{}$, $\sqrt{}$) | W8A8* | PTQ | 6/8 |
| GPTQ (FRANTAR ET AL., 2022) | ($\sqrt{}$, $\times$) | W4 | PTQ + DC | 6/8 |
| SMOOTHQUANT (XIAO ET AL., 2022) | ($\sqrt{}$, $\sqrt{}$) | W8A8 | PTQ + DC | 6/8 |
| OURS | ($\sqrt{}$, $\sqrt{}$) | W6A6/W4A4 | PTQ/TAQ | 8/8 |

Table 1: A comparison of different LLM quantisation methods. (QW, QAct) shows whether quantisations are applied to weights or activations, W$x$A$y$ means $x$-bit quantisation for weights and $y$-bit quantisation for activation. PTQ and TAQ represents Post Training Quantisation and Training After Quantisation respectively. DC means data calibration. There are eight general matrix multiplications (GEMMs) per transformer layer (①-⑧ in Algorithm 2). Only ZeroQuant and ours quantise all of them. Other approaches leave ④ and ⑤ in `float32/float16` format, which take up 20.6% floating-point operations in OPT-6.7B's self-attention. * means outliers in `LLM.INT8()` is computed in `float16`; this improves arithmetic density but memory density is kept identical to canonical `float16`.

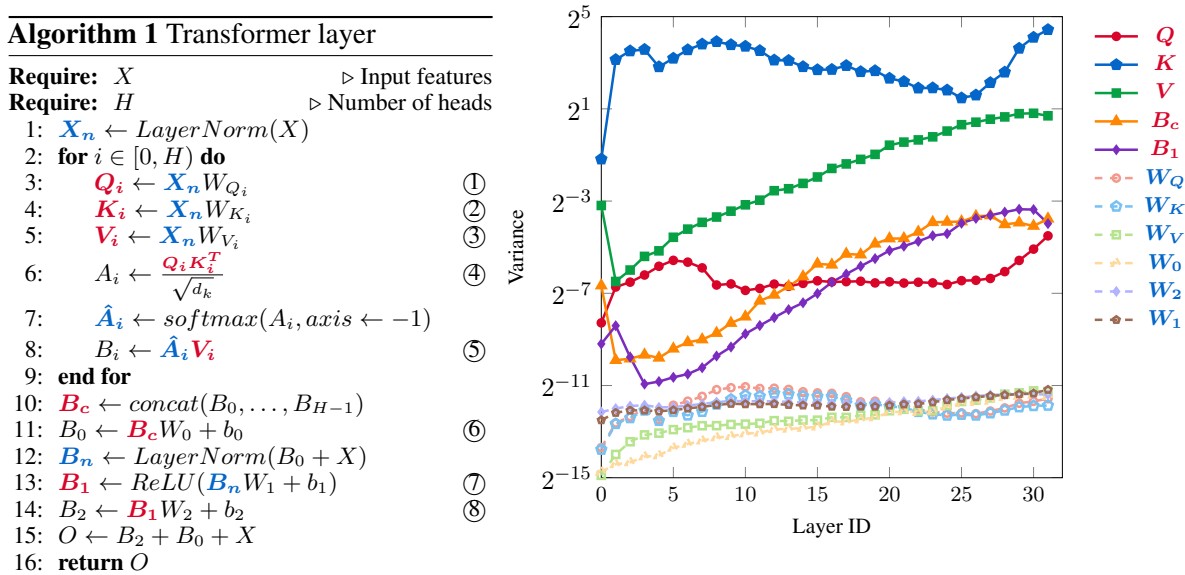

**Algorithm 1** Transformer layer

**Require:** $X$ ▷ Input features
**Require:** $H$ ▷ Number of heads
1: $X_n \leftarrow LayerNorm(X)$
2: **for** $i \in [0, H)$ **do**
3:      $Q_i \leftarrow X_n W_{Q_i}$      ①
4:      $K_i \leftarrow X_n W_{K_i}$      ②
5:      $V_i \leftarrow X_n W_{V_i}$      ③
6:      $A_i \leftarrow \frac{Q_i K_i^T}{\sqrt{d_k}}$      ④
7:      $\hat{A}_i \leftarrow softmax(A_i, axis \leftarrow -1)$
8:      $B_i \leftarrow \hat{A}_i V_i$      ⑤
9: **end for**
10: $B_c \leftarrow concat(B_0, \ldots, B_{H-1})$
11: $B_0 \leftarrow B_c W_0 + b_0$      ⑥
12: $B_n \leftarrow LayerNorm(B_0 + X)$
13: $B_1 \leftarrow ReLU(B_n W_1 + b_1)$      ⑦
14: $B_2 \leftarrow B_1 W_2 + b_2$      ⑧
15: $O \leftarrow B_2 + B_0 + X$
16: **return** $O$

Figure 1: The algorithm on the left is the forward pass computation of a single Transformer layer (Vaswani et al., 2017) in mainstream LLMs, wherein values in blue (*e.g.* $X_n$) represent tensors with predetermined min-max values, such as the outputs of a normalisation layer or softmax. Values in red have unbounded min-max, and are plotted on the upper right for different layers of OPT-6.7B (Zhang et al., 2022). We show that for almost all activation tensors, their variances increase at deeper layers, resulting in *scaling offsets* in their quantisation, while weight tensors on the lower right have smaller variances. This statistical trend enlightens our LLM quantisation study.

2022). After feeding 128 samples from Wikitext2 to the pretrained `float32` model, we make three interesting observations. 1) The variance of most activations in Figure 1 increases with the depth of the layer; 2) Certain tensors (*e.g.* **K**) consistently have a greater variance compared to others; 3) All the weight variance is smaller than activations. Similar trends can be observed in other LLMs. We provide a variance plot of Vicuna-7B (Zheng et al., 2023) in Appendix (Figure 4).

The presence of varying numerical ranges across layers and tensors poses a challenge to the efficacy of a single quantisation configuration for the en-tire network. From an arithmetic perspective, we refer to this phenomenon as *numerical scaling offsets*, as it requires different numerical ranges and granularities for quantisation. To ensure optimal performance, these layers should be subjected to fine-grained non-linear quantisation strategies.

Table 1 provides a comparison between our work and existing LLM quantisation methods. Our quantisation considers all GEMMs (8/8) in transformer layers and both Post-Training-Quantisation (PTQ) and Training-After-Quatisation (TAQ) scenarios. In this work, we also explore suitable places to perform TAQ and quantisation search within the

entire NLP pipeline. We make the following contributions:

- We address the LLM quantisation problem with activation outliers and examine it as a *scaling offsets* problem from an arithmetic design perspective. We demonstrate the efficacy of a family of arithmetic systems with short exponents shared across a block of numbers.

- We propose a novel quantisation framework based on block arithmetic, and demonstrate its effectiveness in performing W6A6 inference for various tasks. Our nearly-lossless W6A6 outperforms prior work in terms of arithmetic density and memory density, without requiring data calibration or fine-tuning.

- We present two methods to achieve 4-bit quantisation on downstream tasks: one is fine-tuning-based, and the other is mixed-precision search. The latter further demonstrates the potential advantage of shifting LLM inference to cost-effective ASICs.

## 2 Related Work

While quantisation of earlier Machine learning (ML) models has been extensively studied, effective quantisation of LLMs still remains an open problem. In this section, we review the previous works on block-based quantisation and compare to the existing LLM quantisation techniques.

### 2.1 Block-based Quantisation

Block-based quantisation is a technique that quantises a block of values into a compact format, where the elements within each block share common digits. This technique offers a significant memory footprint reduction while maintaining a minor round-off error. A number of previous works rely on this method to quantise Convolutional Neural Networks (CNNs). Lin *et al.* utilised a linear combination of multiple binary bases, equivalent to each binary matrix having a scaling factor (Lin et al., 2017). Subsequently, Zhang *et al.* introduced LQ-Nets that rely on a form of block quantisation with a shared scaling factor at the vector level (Zhang et al., 2018). Further investigations explored grouping numbers at various granularities, including layer-wise (Wu et al., 2018b), channel-wise (Krishnamoorthi, 2018), and vector-wise quantisation (Dai et al., 2021).

It is worth noting that sharing a scaling factor is similar to, but not necessarily the same as, sharing the exponent (Darvish Rouhani et al., 2020). This distinction arises because scaling factors can be arbitrary `float32` values, whereas exponent values must be integers represented by the assigned number of bits. Our work focuses on sharing the exponent or exponent bias. When the block size of the shared exponent is 1, we fall back to the minifloat representation such as `FP8` (Sun et al., 2019). These approaches showed promising results primarily for vision models or relatively small Transformer-based models, while we shift the focus to quantising LLMs with a significantly larger parameter count.

### 2.2 LLM Quantisation

Efficient quantisation techniques for language models have been explored in previous works. Zafrir *et al.* proposed an approach for quantising BERT (Shen et al., 2019) into 8-bit integers (Zafrir et al., 2019), while Shen *et al.* (Shen et al., 2019) proposed Hessian-based ultra-low precision quantisation for the same model. Zhang *et al.* (Zhang et al., 2020) quantised BERT to ternary values leveraging layer-wise knowledge distillation, and Bai *et al.* (Bai et al., 2021) further pushed the quantisation of BERT weights to binary values.

The recent surge of interest in quantising LLMs has presented a unique challenge distinct from the prior art summarised above. This challenge stems from the increased model sizes of LLMs. Yao *et al.* proposed ZeroQuant, which quantises both weights and activations of large transformers into small integers with shared scaling factors (Yao et al., 2022). However, as mentioned by Xiao et al. (2022), ZeroQuant suffers from a severe accuracy loss. Dettmers *et al.* introduced `LLM.int8()`, a method that computes outlier GEMMs in `float16` and the rest in 8-bit integer (Dettmers et al., 2022). Xiao *et al.* extended 8-bit LLM quantisation with their PTQ technique named SmoothQuant, Xiao *et al.* proposed SmoothQuant which scales down activations by row and scales up weights by column proportionally before 8-bit fixed-point quantisation (Xiao et al., 2022). Frantar *et al.* proposed GPTQ, which quantises the weights of LLMs to 3 or 4-bit integers while keeping the activations in `float32`. Most LLM quantisation methods, directly or indirectly, reserve LLM activation outliers.

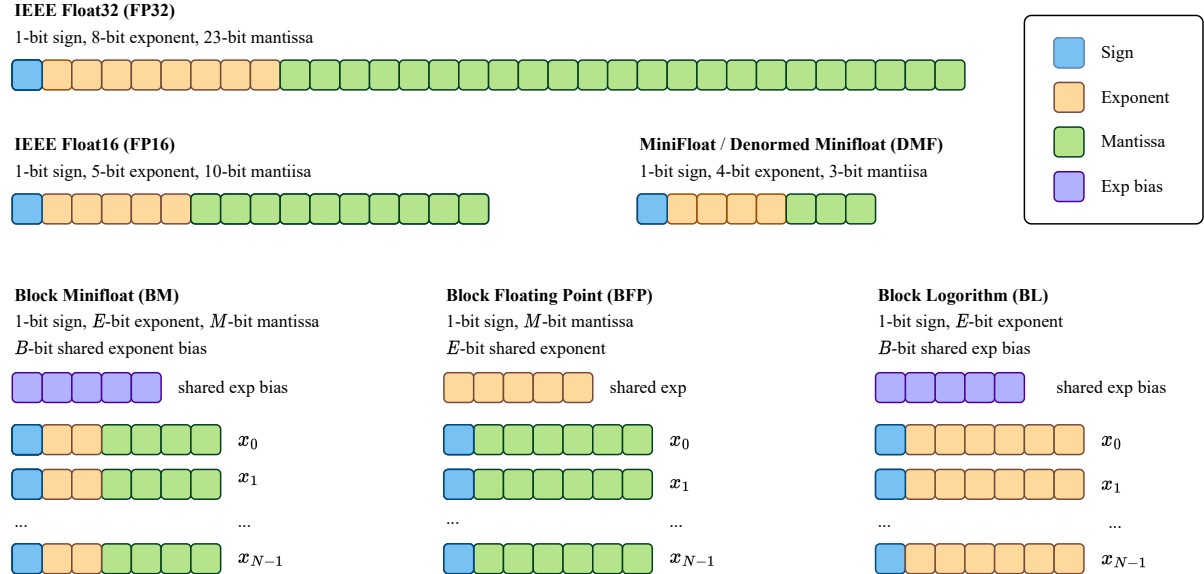

Figure 2: An illustration of different quantisation methods considered in this work: MiniFloat (Sun et al., 2019) and Denormed MiniFloat (DMF), Block MiniFloat (BM) (Fox et al., 2021), Block Floating-Point (BFP) (Darvish Rouhani et al., 2020) and Block Logarithm (BL).

## 3 Method

In this section, we outline our quantisation strategy for LLMs. We first define block-based quantisation and then describe the metrics we use for evaluating quantisation methods. Finally, we detail a precision search that lowers the quantisation granularity down to the tensor level, effectively accommodating the statistical distribution inherent in LLMs.

### 3.1 Block-based Arithmetic

Figure 2 illustrates the data representation we explore to address LLM quantisation as well as the standard `float32`/`float16`. We outline the specifications for traditional floating-point numbers and extend them to block-based quantisation. Detailed definitions can be found in Appendix C.

**Standard floating-point** A standard IEEE floating-point number is defined as a 4-tuple, $(s, e, m, b)$ (Kahan, 1996). $s \in \{0, 1\}$ is the sign bit, $e \in \mathbb{N}$ is the exponent field; $b \in \mathbb{N}$ is the exponent bias; and $m \in \mathbb{N}$ is the mantissa. Let the bit widths of the exponent and the mantissa be $E$ and $M$, respectively. The IEEE standard `float32` (FP32) number has $E = 8$ and $M = 23$, where the other bit is used as the sign bit. Note that the exponent bias depends on $E$: $b = 2^{E-1} - 1$, separating the exponent field symmetrically. Similarly, `float16` (FP16) has $E = 5$ and $M = 10$.

**MiniFloat and Denormalised MiniFloat** MiniFloat is an efficient floating-point representation that requires fewer bits than traditional floating-point numbers. Traditionally, an 8-bit MiniFloat inherits the definition of FP32 by assigning $E = 4$ and $M = 3$. We saturate MiniFloat when $e = 2^E - 1$ and thus no $\pm \inf$ is included.

In this paper, we also introduce a Denormalised MiniFloat (DMF) with zero as the implicit leading bit in the mantissa. Similar to MiniFloat, we saturate the infinity to a maximum finite value. DMF provides a higher precision than MiniFloat for small values at the expense of narrowing down the value range. We investigate this trade-off in the context of quantising LLMs.

**Block MiniFloat, Block Floating-Point and Block Logarithm** As shown in Figure 2, Block quantisation packs values in a block in which a common scaling factor is shared across $N$ values where $N$ is the block size, reducing the computation in vector inner products. This work mainly explores three block quantisation arithmetics on LLMs: BM, BFP and BL.

Block Minifloat (BM) shares a $B$-bit exponent bias (Fox et al., 2021). This representation achieves high precision and high range at the same time, at the cost of a larger quantisation error at medium value than standard floating point. This is potentially amenable to values in a multimodal distribu-

tion, where values close to a peak can be efficiently represented in a block. Block Floating-Point (BFP) shares an $E$-bit exponent. This shared exponent bounds the range in the block and is amenable to values with small block variances. Block Logarithm (BL) sets the mantissa in BM to 1 and shares a $B$-bit exponent bias, resulting in values that are powers-of-twos. This contrasts with BFP and is amenable to values with large dynamic ranges.

All these quantisation methods are non-linear and thus can be useful tools to address the *scaling offsets* phenomenon depicted in Figure 1. Moreover, the hyper-parameter block size allows for flexible quantisation granularity, ranging from layer-wise, tensor-wise, and channel-wise, to slice-wise (a slice along the token/channel vector).

## 3.2 Arithmetic and Memory Densities

Reducing model size is not the only advantage of quantisation; it also simplifies the computation, thereby accelerating inference. We evaluate quantisation arithmetics using adopted memory and arithmetic densities (Darvish Rouhani et al., 2020). We define memory density as the reciprocal of the size of the activation and weight data in a model, and the arithmetic density as the reciprocal of the area/the number of Look-Up-Tables (LUTs) to synthesise a multiply-accumulate (MAC) unit, which serves as the basic cell for matrix multiplication in custom inference circuits. An efficient quantisation method should make a good trade-off among task accuracy, memory density, and arithmetic density. We implemented MAC units with different above-mentioned arithmetics in FPGAs to obtain the number of LUTs. A detailed description of this procedure can be found in Appendix D.

## 3.3 Quantisation Search

Previous works (Dong et al., 2019; Habi et al., 2020) observed that the layers in CNNs exhibit varying tolerance, or "sensitivity", to quantisation – we also notice this phenomenon in LLMs. The crucial aspect is identifying the layers that are sensitive and determining tailored quantisation configurations. To achieve this, we apply Tree-structured Parzen Estimator (TPE) (Bergstra et al., 2011) to conduct a fine-grained search for quantisation precision multiple times and analyse the statistics inherent in the quantised models that recover more accuracy. Our search space is constructed on a per-tensor basis, allowing each input tensor or weight tensor in ①-⑧ (See Algorithm 2) to have its own

| Method | Config | $E$ | $M$ | $B$ |
|---|---|---|---|---|
| Fixed-point | W8A8 | - | 7 | - |
| MiniFloat | W8A8 | 4 | 3 | - |
| DMF | W8A8 | 4 | 3 | - |
| BFP | W8A8 | 8 | 7 | - |
| BFP | W6A6 | 8 | 5 | - |
| BFP | W4A4 | 8 | 3 | - |
| BM | W8A8 | 4 | 3 | 8 |
| BL | W8A8 | 7 | - | 8 |

Table 2: The quantisation configuration used in the following sections, where $E$, $M$, and $B$ are the bit-width of exponent (shared exponent), mantissa, and bias (shared bias) respectively.

precision. The search space increase exponentially as the layer count increases. We leverage accuracy and memory density to design the objective function: $O_f = acc + \alpha \cdot mem$. Here $O_f$, $acc$, $mem$ represent the objective function, accuracy, and memory density of the searched quantised models, respectively. The constant $\alpha$ is used to balance $acc$ and $mem$. To determine the $\alpha$ for a specific search, we initially set $\alpha$ to 1.0 and perform the search while recording the values of $(acc, mem)$ until convergence. The final value of $\alpha$ is determined as $\frac{acc_c}{mem_c}$, where $(acc_c, mem_c)$ represents the converged values. Detailed search parameters are in Appendix B.

## 4 Evaluation

We conducted a comprehensive set of experiments to identify the key factors influencing the performance of sub-8-bit LLMs. We begin with a language modelling task to eliminate less promising quantisation methods (Section 4.2), and then run the promising ones on downstream tasks. For the tasks that proved challenging even for FP32 models, we resort to fine-tuning. Additionally, we conducted a mixed-precision search on two tasks where the quantised 4-bit model struggle. The results of this search provide insights into how to further refine quantisation at the tensor level.

## 4.1 Experiment setup

**Baselines** We compare our approach with four baselines: 8-bit plain fixed-point quantisation, `LLM.int8()` (Dettmers et al., 2022), GPTQ (Frantar et al., 2022), and SmoothQuant (Xiao et al., 2022). We amend SmoothQuant's source code to ensure its consistency with their paper (See Ap-

| Method | Config | Perplexity (↓) | | | | | Hardware metrics | |
|---|---|---|---|---|---|---|---|---|
| | | 125M | 350M | 1.3B | 2.7B | 6.7B | Mem ↑ | Arith ↑ |
| FP32 | - | 27.65 | 22.00 | 14.62 | 12.47 | 10.86 | 1× | 1× |
| LLM.int8() | W8A8† | 27.72 | 22.03 | 14.64 | 12.49 | 10.86 | 2× | < 7.7× |
| GPTQ | W4* | 31.12 | 24.24 | 15.47 | 12.87 | 11.39 | < 1.6× | - |
| SmoothQuant | W8A8 | -‡ | -‡ | 14.62 | 12.50 | 10.85 | < 4× | < 7.7× |
| SmoothQuant-c | W8A8 | -‡ | -‡ | 17.97 | 26.88 | 42.90 | 4× | 7.7× |
| Fixed-point | W8A8 | 275 | 117 | 1.78E4 | 7.81E3 | 3.77E3 | 4× | 7.7× |
| MiniFloat | W8A8 | 28.16 | 22.24 | 15.03 | 12.73 | 10.99 | 4× | 17.4× |
| DMF | W8A8 | 30.41 | 23.89 | 18.08 | 14.55 | 11.95 | 4× | 17.4× |
| BFP | W6A6 | **28.27** | **22.22** | **15.08** | **12.54** | **10.90** | **4.9×** | **19.2×** |
| BFP | W4A4 | 41.94 | 33.98 | 24.70 | 19.34 | 13.59 | 7.1× | 37.3× |
| BM | W8A8 | 5.6E3 | 2.7E4 | 1.17E4 | 1.33E4 | 8.61E3 | 3.8× | 14.4× |
| BL | W8A8 | 780 | 1.26E3 | 323 | 950 | 289 | 3.8× | 16.1× |

Table 3: Perplexity (↓) values with zero-shot Post-Training-Quantisation (PTQ) on WikiText2, this means we directly quantise the pre-trained model and apply on WikiText2. Mem and Airth represent Memory and Arithmetic density accordingly. DMF, BM, BFP and BL represent Denormalised MiniFloat, Block Minifloat, Block Floating Point and Block Logarithm respectively. SmoothQuant-c is our improved implementation where the two activation matrix multiplications are now also quantised. † means the inliner matrix multiplications are calculated in 8-bit fixed-point, and outliers are calculated in FP16. * means the weights of GPTQ are kept in FP32. ‡ means SmoothQuant repository does not include the weight scaling matrices for 125M and 350M. We **highlight** the best block-based quantisation arithmetic, 6-bit BFP, considering perplexity, memory density, and arithmetic density together.

| Model | FP32 | LLM.int8() | W6A6 BFP |
|---|---|---|---|
| LLaMA-7B | 5.79 | 5.83 (+0.04) | 5.83 (+0.04) |
| Vicuna-7B | 7.06 | **7.07 (+0.01)** | 7.08 (+0.02) |
| Alpaca-7B | 7.01 | 7.02 (+0.01) | 7.02 (+0.01) |
| LLaMA-13B | 5.17 | 5.22 (+0.05) | **5.20 (+0.03)** |
| Vicuna-v1.5-13B | 6.13 | 6.16 (+0.03) | 6.16 (+0.03) |

Table 4: Perplexity (↓) values of LLM family quantized by W6A6 BFP. We compare our method with FP32 and LLM.int8() and find that our method achieves nearly lossless perplexity on Wikitext2. We exclude GPTQ and SmoothQuant-c in this table because they have obvious perplexity increase larger than 0.2 and 5.0 respectively.

pendix B) and add this amended version (referred to as "SmoothQuant-c") to the result table.

**Quantisation configuration** Table 2 clarifies the quantisation configuration used in the following sections, where $E$, $M$, and $B$ are the bit-width of exponent (shared exponent), mantissa, and bias (shared bias) respectively. All these representations include a 1-bit sign bit. The block size of block-based methods is set to $[1, 16]$ for both the weight and activation matrix (a slice along matrix row in Algorithm 2) unless otherwise specified.

**Models and datasets** We choose the representative OPT (Zhang et al., 2022) family, and evaluate on Wikitext2 (Merity et al., 2016), ARC(easy) (Clark et al., 2018), LAMBADA (Paperno et al., 2016), PIQA (Bisk et al., 2020), COPA (Roemmele et al., 2011), QNLI (Wang et al., 2018), SST2 (Socher et al., 2013), MRPC (Dolan and Brockett, 2005), and COLA (Warstadt et al., 2019). To demonstrate the generalizability of our method, we also report the Wikitext2 perplexity of quantized LLaMA models (Touvron et al., 2023; Chiang et al., 2023; Taori et al., 2023). Following prior work (Zhang et al., 2022; Xiao et al., 2022), we use lm-eval-harness (Gao et al., 2021) to evaluate models on downstream tasks in the context of zero-shot prompting.

## 4.2 Zero-shot PTQ on Wikitext2 and downstream tasks

In this section we present our results in a setup we call zero-shot Post-Training-Quantisation (PTQ), which was also adopted by prior work on LLM quantisation (Dettmers et al., 2022; Frantar et al., 2022; Xiao et al., 2022). In this approach, we take a pre-trained OPT model from Huggingface, quantise it, and apply it on Wikitext2 to calculate

| Method | Config | Mean accuracy (↑,%) | | | | |
|--------|--------|------|------|------|------|------|
| | | 125M | 350M | 1.3B | 2.7B | 6.7B |
| Float32 | - | 52.7 | 57.5 | 69.6 | 65.4 | 73.4 |
| LLM.int8() | W8A8 | 52.5 (-0.2) | 58.3 (+0.8) | 69.2 (-0.4) | 65.3 (-0.1) | 73.5 (+0.1) |
| LLM.int4() | W4A4 | 50.8 (-1.9) | 55.8 (-1.7) | 67.0 (-2.6) | 64.5 (-0.9) | 72.5 (-0.9) |
| SmoothQuant-c | W8A8 | - | - | 67.2 (-2.4) | 65.2 (-0.2) | 72.2 (-1.2) |
| MiniFloat | W8A8 | 52.1(-0.6) | 55.1(-2.4) | 64.7(-4.9) | 65.7(+0.3) | 70.5(-2.9) |
| BFP | W4A4 | 47.8 (-4.9) | 51.7 (-5.8) | 57.2 (-12.4) | 55.7 (-9.7) | 67.2 (-6.2) |
| BFP | W5A5 | 51.1 (-1.6) | 56.8 (-0.7) | 65.5 (-4.1) | 64.6 (-0.8) | 72.0 (-1.4) |
| BFP | W6A6 | **52.6 (-0.1)** | **57.6 (+0.1)** | **67.8 (-1.8)** | **65.5 (+0.1)** | **72.9 (-0.5)** |
| BFP | W8A8 | 52.8 (+0.1) | 57.6 (+0.2) | 69.1 (-0.5) | 65.2 (-0.2) | 73.1 (-0.3) |

Table 5: Mean accuracy (↑, %) values with zero-shot prompting PTQ on ARC (easy), COPA, LAMBADA, PIQA, and SST2, this means we directly quantise the pre-trained model and benchmark on these downstream tasks using zero-shot prompting. We **highlight** 6-bit BFP which also achieves an accuracy close to FP32 on these tasks.

perplexity, and the eight downstream tasks short-listed in Section 4.1 to calculate accuracy. The zero-shot PTQ setup is particularly advantageous in scenarios where LLMs lack prior knowledge, as it eliminates the need for downstream task fine-tuning and Training-After-Quantisation (TAQ).

**Perplexity on Wikitext2**  Table 3 compares our results with the baselines in terms of perplexity, memory density, and arithmetic density. Similar to prior work (Dettmers et al., 2022; Xiao et al., 2022), plain fixed-point quantisation performs poorly. In contrast, non-linear arithmetic, such as MiniFloat, yields a significantly better perplexity at a similar memory density. MiniFloat yields slightly better results than DMF, indicating the $2\times$ higher range is more important than precision in this context.

Block-based quantisation exhibits inconsistent performance on Wikitext2. A noteworthy result is that our 6-bit BFP achieves higher memory density, higher arithmetic density, and lower perplexity than the prior art GPTQ and SmoothQuant-c without requiring data calibration. BM and BL perform poorly compared to BFP. BM was originally proposed in the context of Quantisation-Aware-Training (QAT), whereas our evaluation is based on PTQ. Without retraining, the 3-bit mantissa of BM and the 1-bit mantissa of BL may be the reason for the poor perplexity.

Table 4 shows the perplexity of W6A6 BFP on LLaMA family, including LLaMA-7B/-13B (Touvron et al., 2023), Vicuna-7B (Zheng et al., 2023), Alpaca-7B (Chiang et al., 2023), and Vicuna-v1.5-13B (Chiang et al., 2023), with FP32 and

LLM.int8() as baselines. We observe that 6-bit BFP still achieves nearly lossless perplexity on these models, verifying the efficacy of our method across model architectures.

**Accuracy on downstream tasks**  We exclude fixed-point, DMF, BM, and BL from downstream task evaluation due to their poor language modelling performance. Table 5 represents the mean accuracy on ARC (easy), COPA, LAMBADA, PIQA, and SST2. The results of QNLI, MRPC, and COLA are not included in this table as even FP32 LLMs exhibited poor accuracy close to random guess. A plot depicting how these methods match FP32 accuracy as the model scales up and a complete result table are in Appendix E.

Besides LLM.int8() and SmoothQuant-c, we also report a 4-bit version LLM.int8() (referred to as LLM.int4()) reported by Dettmers (2023) on downstream tasks. We observe that 6-bit BFP achieve nearly lossless accuracy, below FP32 and LLM.int8(), and above SmoothQuant-c and LLM.int4(). Note that 6-bit BFP has the highest memory density and arithmetic density among these methods. The 4-bit BFP suffers severe accuracy degradation because its shared exponent and 3-bit mantissa cause large quantisation errors.

Overall, we make the following observations:

- Fixed-point representation performs inadequately due to unability of linear quantisation to address the scaling offset issue caused by varying variances.

- LLMs have different tolerance to block-based

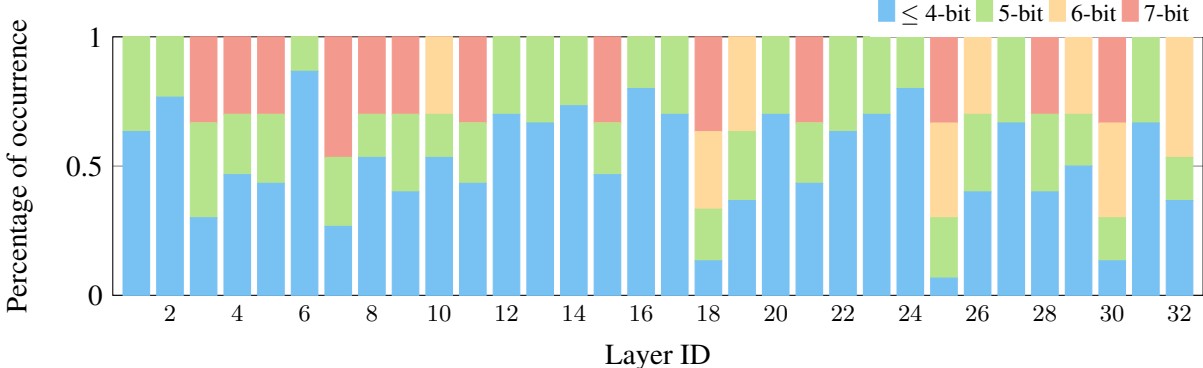

Figure 3: The bit width distribution of **Q** in Line 6, Algorithm 2 from 2688 searches. We identify the layers less tolerant to aggressive quantisation in OPT-2.7B. For example, layers 18, 25 and 30 often need more bits than other layers. Keeping these layers in relatively high precision recovers the accuracy from 36.2% to 61.3% without decreasing the memory density, equivalent to a 4.3-bit OPT-2.7B on average.

quantisations. BM and BL exhibit subpar performance compared to BFP, indicating that non-linear quantisation still needs sufficient mantissa length to capture the learned weight distribution, or retraining may be required.

- BFP strikes a good balance in the trade-off between range and resolution. Our nearly-lossless 6-bit LLMs, without data calibration/re-training, outperform prior art methods in terms of perplexity (accuracy), memory density, and arithmetic density.

We also observe that sub-6-bit BFP has a severe accuracy drop. To address this problem, we further investigate two approaches for improving the accuracy of 4-bit LLMs.

### 4.3 4-bit LLMs via fine-tuning

Previous study (Brown et al., 2020; Zhang et al., 2022) reported FP32 LLMs' low accuracy on several downstream tasks in the context of zero-shot prompting. In our experiments, OPTs also exhibit poor accuracy on QNLI, MRPC, and COLA. Fine-tuning language models on downstream tasks has proven to be helpful for improving accuracy (Devlin et al., 2019). We explore the fine-tuning and quantisation of LLMs on downstream tasks.

There are two stages where quantisation can be applied. LLMs are typically pre-trained in FP32. The first option is to continue fine-tuning the FP32 model on downstream tasks and subsequently quantise this fine-tuned FP32 model. We refer to this setup as *PTQ on fine-tuned FP32*. The second option is to quantise the pre-trained FP32 model and

retrain this quantised model on downstream tasks, which we refer to as *TAQ on downstream tasks*.

We compare these two cases on four downstream tasks (SST2, QNLI, MRPC, and COLA) that zero-shot prompting struggles to handle. The result table is in Appendix F. We observe that:

- Both options effectively improve accuracy, enabling nearly lossless downstream accuracy even if 4-bit BFP is applied.

- TAQ on downstream tasks reaches a slightly better accuracy (a gain of 0.2% on average) than PTQ on fine-tuned FP32 given the same bit-width. However, the former is harder to optimize through backpropagation because of the forward quantisation error and the Straight-Through Estimator (STE) (Bengio et al., 2013) used in backpropagation.

### 4.4 4-bit LLMs via mixed precision

Currently, our block-based quantisation uses a uniform configuration, where the block size and bit-width remain constant across the entire model. What if we push the barrier further? Existing works on CNN compression have explored mixed-precision quantisation (Wu et al., 2018a; Wang et al., 2019), thereby increasing memory density. This subsection lowers the block size granularity and the bit-width granularity to the tensor level to demonstrate uncharted possibilities of aggressive LLM quantisation.

**Variation-aware block size** By comparing the activation variance and weight variance in Figure 1, we observe that the weight variance remains stable

and much smaller, suggesting that we can increase the weight block size while decreasing the activation block size. This approach enhances accuracy while maintaining memory density.

**Mixed-precision** We repeat the quantisation search described in Section 3.3 on downstream tasks and filter out less promising quantisation configurations using an accuracy threshold and a memory density threshold. Each time we start TPE search with a different random seed, so the distribution of filtered quantisation configurations exposed the sensitivity of the searched tensors in LLMs. An example of a mixed-precision search result is presented in Figure 3. We find *certain layers were consistently assigned with higher precision, while others tended to have lower bit widths*. By preserving high precision for these sensitive layers, we recovered the 4-bit LLM accuracy *from 36.2% to 61.3%* on LAMBADA without compromising memory density. The memory density of the searched OPT-2.7B is $7.42\times$, which is slightly better than the uniform 4-bit BFP's $7.11\times$. Figure 7 in Appendix G compares uniform 4-bit BFP and mixed-precision 4-bit BFP on LAMBADA and ARC (easy), highlighting the effectiveness of our mixed-precision quantisation. We include more tasks and model sizes in Appendix G. In conclusion, variance-aware block size and mixed precision allow aggressive quantisation beyond 6-bit without fine-tuning.

## 5 Conclusion

This study focuses on addressing the scaling offset issue in LLMs and provides valuable insights into the quantisation of LLMs. Through extensive experimentation, we identify key factors that significantly impact LLM quantisation. When aiming for quantisation above or equal to 6-bit, BFP surpasses previous methods in terms of accuracy, memory density, and arithmetic density, without requiring for data calibration or training. Moreover, we demonstrate that fine-tuning or mixed precision techniques enable 4-bit LLMs on downstream tasks. Fine-tuning is suitable for GPUs, and mixed precision has the potential to shift the inference platform from GPUs to cost-effective ASICs. Our findings contribute to advancing the field of LLM quantisation and provide practical guidance for achieving good quantisation performance. Our work has been open-sourced and will also contribute to an ML hardware framework named MASE for hardware deployment (Cheng et al., 2023).

## Limitations

Different from many prior arts in LLM quantisation that focus on integers, our work puts particular emphasis on minifloat variants. However, the potential gains of our work have not manifested in GPU systems due to a lack of CUDA kernel implementation. The implementation of some proposed quantisation methods in this paper requires specialised kernels and hardware, however, a major focus of our work is to *explore potential designs for next-generation hardware to run LLM inference*. Another limitation is that our search algorithm does not include arithmetic density due to a lack of hardware models for LLMs. We ran a mixed-precision search with hardware models on a small transformer. The result included in Appendix G is promising. We leave sufficient study on hardware-aware LLM quantization as a future work.

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

# A   Tensor variance in LLMs

We investigate the variance trend of Vicuna-7B (Chiang et al., 2023) and observe the same trend of increasing activation variances as OPT-6.7B. Figure 4 depicts the forward pass of Vicuna-7B and the variance trend of its tensors. Interestingly, the variances of self-attention input activations ($\mathbf{Q}'_i$ and $\mathbf{K}'_i$ in line 6 of the algorithm above) are consistently high from the first Transformer layer to the last in Vicuna-7B. We assume this is because of the Rotary Positional Encoding (RoPE) (Su et al., 2021) layers.

We additionally analyse the trend of increasing activation variance when the model size increases,

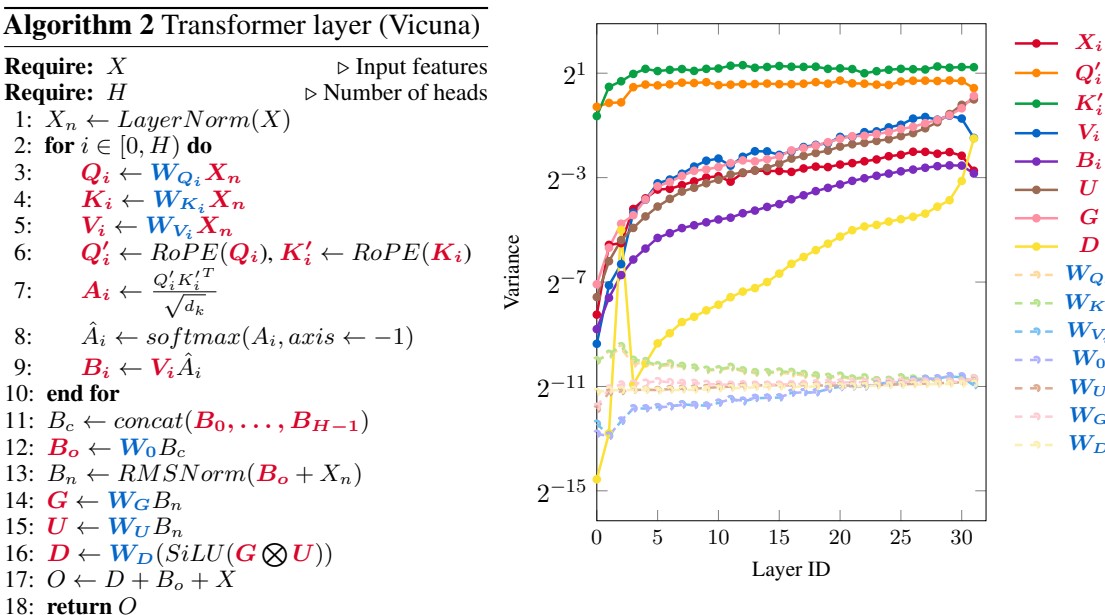

**Algorithm 2** Transformer layer (Vicuna)

**Require:** $X$        ▷ Input features
**Require:** $H$        ▷ Number of heads
1:   $X_n \leftarrow LayerNorm(X)$
2:   **for** $i \in [0, H)$ **do**
3:      $Q_i \leftarrow W_{Q_i} X_n$
4:      $K_i \leftarrow W_{K_i} X_n$
5:      $V_i \leftarrow W_{V_i} X_n$
6:      $Q_i' \leftarrow RoPE(Q_i), K_i' \leftarrow RoPE(K_i)$
7:      $A_i \leftarrow \frac{Q_i' K_i'^T}{\sqrt{d_k}}$
8:      $\hat{A}_i \leftarrow softmax(A_i, axis \leftarrow -1)$
9:      $B_i \leftarrow V_i \hat{A}_i$
10: **end for**
11: $B_c \leftarrow concat(B_0, \dots, B_{H-1})$
12: $B_o \leftarrow W_0 B_c$
13: $B_n \leftarrow RMSNorm(B_o + X_n)$
14: $G \leftarrow W_G B_n$
15: $U \leftarrow W_U B_n$
16: $D \leftarrow W_D(SiLU(G \otimes U))$
17: $O \leftarrow D + B_o + X$
18: **return** $O$

Figure 4: The algorithm on the left is the forward pass of Vicuna-7B. The graph on the right depicts how the tensor variances change with layer number. We observe the same trend of increasing activation variances in Vicuna-7B, which is similar to OPT-6.7B. interestingly, the variances of self-attention input activations ($Q_i'$ and $K_i'$ in line 6 of the algorithm above) are consistently high from the first Transformer layer to the last in Vicuna-7B. We assume this is because of the Rotary Positional Encoding (RoPE) (Su et al., 2021) layers.

In Figure 1, for OPT-6.7B, we plotted the variances of all tensors that have unbounded input ranges and that are taken as input operands to matrix multiplications in the Transformer layer. Figure 5 further illustrates the results for OPT-350M and OPT-2.7B. We observe that:

- If we consider $V$, $B_c$ and $B_1$ as the *main information path* [2], these components have much smaller variances than $K$ and $Q$.

- Bigger models tend to have small variances at shallow layers and larger variances at deep layers.

These observations explain why linear quantisation, such as integer quantisation, is effective for smaller models but struggles with larger ones. This increasing activation variance trend can be considered into variance-aware block size. Since a higher variance implies a higher possibility of extreme outliers, we can apply larger block sizes to those tensors with smaller variance and smaller block sizes to those with higher variance. Limited by time, we leave this exploration as well as the combination of fine-tuning, variance-aware block size, and mixed precision in future work.

---

[2] The computation of $Q$ and $K$ yields attention factors (post-softmax) that are applied to $V$.

## B   Experiment details

### B.1   Setup and Implementation

**Hardware resources**   We run the experiments using four NVIDIA RTX3090s, three A100s, and eight V100s with 64GB, 192GB, and 128GB RAM respectively. The evaluation of PTQ perplexity on Wikitext2 takes around 64 GPU hours in total; the zero-shot prompting evaluation on downstream tasks takes around 160 GPU hours in total; the fine-tuning of FP32 models on SST2, QNLI, MRPC and COLA takes around 30 GPU hours in total; the fine-tuning of quantised BFP models takes around 70 GPU hours in total; the evaluation of fine-tuned models takes around 6 GPU hours in total; the mixed-precision search takes around 120 GPU hours in total.

**Implementation**   We download the model codes and pre-trained weights from HuggingFace Transformers [3] and implement the quantisation arithmetics using PyTorch [4]. We use Vivado to report arithmetic density and Optuna[5] to perform the mixed-precision search.

---

[3] https://github.com/huggingface/transformers
[4] https://github.com/pytorch/pytorch
[5] https://optuna.readthedocs.io/en/stable/index.html

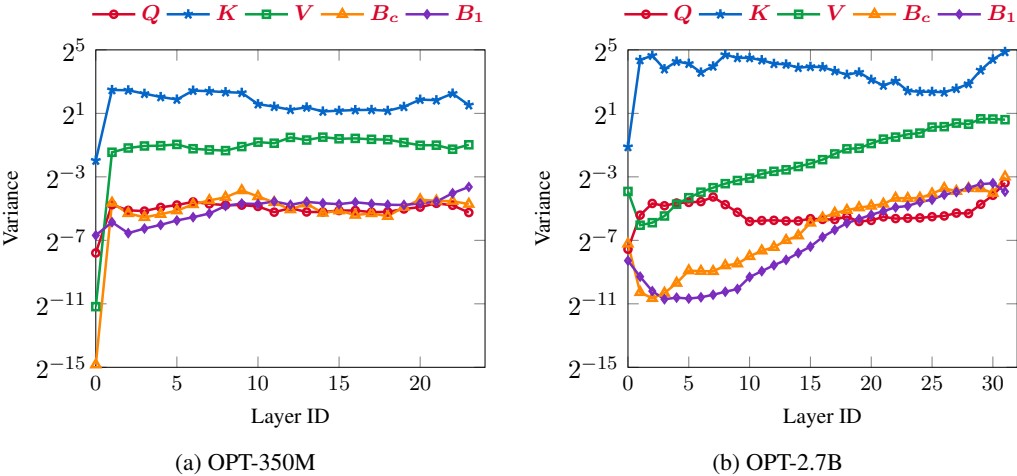

(a) OPT-350M            (b) OPT-2.7B

Figure 5: We demonstrate a similar analysis to Figure 1, where on the left we have OPT-350M variance vs layer ID and OPT-2.7B variance vs layer ID on the right. The trend of increasing activation variance is more obvious on larger models.

**Evaluation** We follow the code base of GPTQ (Frantar et al., 2022)[6] to estimate LLM's perplexity on Wikitext2. We chop Wikitext2's test set into sequences of 2000 tokens, feed the sequences to LLMs, and normalise the cross entropy loss by the sequence length and batch size. To evaluate LLM accuracy on downstream tasks, we follow OPT (Zhang et al., 2022) and SmoothQuant (Xiao et al., 2022) to use `lm-eval-harness` in the zero-shot prompting setup.

### B.2 Comparison with SmoothQuant

The SmoothQuant paper (Xiao et al., 2022) declares all the eight GEMMs (①-⑧ in 2) are quantised. However, their codes[7] do not support quantising ⑤ and ⑥, which takes up 19.6% floating-point operations (FLOPs) in OPT-6.7B's self-attention. We amend their code and refer to the amended version as "SmoothQuant-c", which should be the same as SmoothQuant-O2 in the paper. We observe that SmoothQuant-c has much higher perplexity and slightly lower accuracy on downstream tasks than SmoothQuant. Besides, the SmoothQuant repository does not include the scaling factor files of OPT-125m and OPT-350m, so the perplexity/accuracy for these two models is missing in our result table.

### B.3 Comparison with `LLM.int8()`

We give a short comparison between `LLM.int8()` and our method. `LLM.int8()` is different from plain 8-bit fixed-point quantization. In `LLM.int8()` all the tensors are stored as FP16 numbers, which is the reason why `LLM.int8()` has 2× memory density while plain 8-bit fixed-point has 4× memory density in Table 3. `LLM.int8()` targets GPUs while ours targets ASICs. `LLM.int8()` is not as friendly as uniform BFP to ASICs, because `LLM.int8()` separates one matrix multiply into two (one for inliers the other for outliers), casts inliers to 8-bit, performs an 8-bit matrix multiplication for inliers and an FP16 matrix multiplication for outliers respectively. This separation is performed on the fly. In comparison, 6-bit uniform BFP does not require a runtime separation or an FP16 matrix multiply engine. All tensors are stored and calculated in 6-bit BFP.

### B.4 Quantisaion search

The specific search configuration depends on model size, task, and FP32 performance. We use the accuracy threshold and memory density threshold to sort out promising mixed-precision configs. Given a model and a task, the accuracy threshold is 2% below the FP32 values. The memory density is set to 7.1% in most search configs.

Note that to estimate the memory density of quantisation config candidates, we need the model architecture information including input sizes and weight sizes for all the GEMMs in Algorithm 2 across all layers. We implement a FLOP profiler to collect this information and feed it as input to the search algorithm. The numeric values of these parameters can be found in the bash scripts of our

---

[6]https://github.com/IST-DASLab/gptq
[7]https://github.com/mit-han-lab/smoothquant

source code.

## C  Definition of quantisation arithmetics

**FP32, FP16 and MiniFloat**  A traditional floating-point representation follows IEEE floating-point standard (Kahan, 1996), which can define a floating-point number as a 4-tuple, $(s, e, m, b)$, where

- $s \in \{0, 1\}$ is the sign bit of the number;

- $e \in \mathbb{N}$ is the exponent field;

- $b \in \mathbb{N}$ is the exponent bias; and

- $m \in \mathbb{N}$ is the mantissa.

Given the bit widths of the exponent and the mantissa be $E$ and $M$, the value $x$ of a floating-point number can be obtained via:

$$x = \begin{cases} (-1)^s \times 2^{1-b} \times \frac{m}{2^M} & e = 0 \\ (-1)^s \times 2^{e-b} \times (1 + \frac{m}{2^M}) & 0 < e < 2^E - 1 \\ (-1)^s \times \infty & e = 2^E - 1, m = 0 \\ \text{NaN} & \text{others} \end{cases}$$
(1)

where $e$ is the unsigned integer value represented by the exponent bits, and $m$ is the unsigned integer value represented by mantissa bits. The exponent bias ($b$) is a constant depending on $E$: $b = 2^{E-1} - 1$. FP32, FP16, and MiniFloat have $E = 8, M = 23$, $E = 5, M = 10$ and $E = 4, M = 3$, respectively. Note that the "1" in the fraction term of Line 2, Equation (1) comes from the implicit leading bit in the mantissa.

We additionally saturate MiniFloat when $e = 2^E - 1$, thus the value of a MiniFloat is

$$x = \begin{cases} (-1)^s \times 2^{1-b} \times \frac{m}{2^M} & e = 0 \\ (-1)^s \times 2^{e-b} \times (1 + \frac{m}{2^M}) & 0 < e \leq 2^E - 1 \\ \text{NaN} & \text{others} \end{cases}$$
(2)

**DMF**  The definition of DMF is the same as MiniFloat except that there is no implicit leading bit in the mantissa:

$$x = \begin{cases} (-1)^s \times 2^{e-b} \times \frac{m}{2^M} & e \leq 2^E - 1 \\ \text{NaN} & \text{others} \end{cases}$$
(3)

**BM, BL, and BFP**  BM (Fox et al., 2021) shares the exponent bias and was proposed in the context of Quantisaion-Aware-Training (QAT). When an FP32 value is cast to BM, the exponent bias is determined by the maximum value in the block.

BFP (Darvish Rouhani et al., 2020) shares the exponent and was proposed in the context of PTQ.

Similar to BM, the shared exponent bias is also determined by the maximum FP32 values when casted from FP32.

Logarithm quantisation was proposed by Miyashita et al. (2016) to perform QAT on CNNs. Block Logarithm (BL) was used as a baseline to compare with BM in (Fox et al., 2021). BL shares the exponent bias and does not have mantissa bits (mantissa is always 1).

Basically, block-based quantisation facilitates the vector's inner product by simplifying the accumulation after multiplication. For example, the inner product between two BFP vectors $\mathbf{x}$ and $\mathbf{y}$ is,

$$\begin{aligned} &\mathbf{x} \cdot \mathbf{y} \\ =& (-1)^{s_x} e^{e_x} [x_1, \ldots, x_{B-1}] \cdot \\ & (-1)^{s_y} e^{e_y} [y_1, \ldots, y_{B-1}] \\ =& (-1)^{s_x + s_y} e^{e_x + e_y} [x_1 y_1 + \cdots + x_{B-1} y_{B-1}] \end{aligned}$$
(4)

where $B$ is the block size. Since exponents are shared across vectors, the element products can be accumulated without shifting. The block sizes of the two vectors are not necessarily the same.

## D  Estimate arithmetic density via logic synthesis

We implemented the hardware designs of the corresponding modules and measured their arithmetic density using hardware synthesis tools. Each design contains versions for Int8, Float32, MiniFloat and all the block arithmetic types above. All these designs are functionally verified in AMD Xilinx simulator on a set of test vectors. The hardware arithmetic density is obtained using the same formulation by Darvish Rouhani *et al.* (Darvish Rouhani et al., 2020) with area in FPGAs. The area results were obtained from the post-Place & Synthesis report in AMD Xilinx Vivado (AMD Xilinx Vivado, 2023). We estimate the total circuit area in LUTs, and a DSP is considered to be equivalent to 100 LUTs. We used the UltraScale+ family of FPGA devices for experiments, and the version of AMD Xilinx software is 2020.2. The arithmetic densities of various quantisation methods are present in Table 6.

## E  PTQ on downstream tasks

We quantised the pre-trained model and apply it to the downstream tasks in the zero-shot prompting setup. Figure 6 depicts how the performance of quantised models scale with model sizes. Our 6-bit BFP align with FP32 at various model sizes.

| Method | Config | Block size | #DSPs | #LUTs | Area factor | Arithmetic density |
|---|---|---|---|---|---|---|
| FP32 | - | 1 | 5 | 335 | 835 | 1× |
| Integer | W8A8 | 1 | 1 | 9 | 109 | 7.7× |
| MiniFloat | W8A8 | 1 | 0 | 48 | 48 | 17.4× |
| BM | W8A8 | 1 | 0 | 27 | 51 | 16.4× |
| BFP | W8A8 | 16 | 0 | 544 | 58 | 14.4× |
| BL | W8A8 | 1 | 0 | 28 | 52 | 16.1× |
| BFP | W6A6 | 16 | 0 | 313 | 43.6 | 19.2× |
| BFP | W4A4 | 16 | 0 | 358 | 22.4 | 37.3× |

Table 6: The arithmetic density of various quantisation configurations explored in this paper. To calculate the area factor, we convert the Digital Signal Processing units (DSPs) to equivalent LUTs to get the area factor, and then divide the quantisation arithmetic's area factor density by FP32.

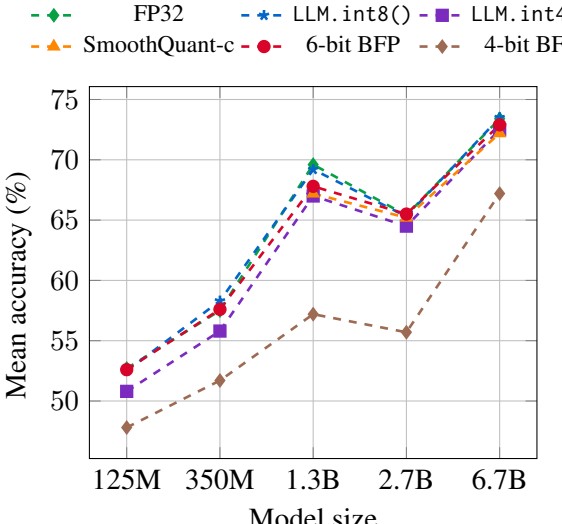

Figure 6: Mean accuracy (↑, %) of various quantisation methods on downstream tasks. We observe that 6-bit BFP align with FP32 as model size scales, above SmoothQuant-c and LLM.int4(), below FP32 and LLM.int8(). Note that 6-bit BFP (- ● -) achieves the highest memory density and arithmetic density among these methods. 4-bit BFP (- ◆ -) has a severe accuracy drop.

Table 7 presents the detailed accuracy of each task. Note that QNLI, MRPC, and COLA results are not included in this table because even FP32 LLMs yield an accuracy close to random prediction.

## F  PTQ on fine-tuned FP32 vs TAQ on downstream tasks

Table 8 compares the two options on four downstream tasks (QNLI, SST2, COLA, MRPC), that FP32 LLM cannot handle. We observe that both align 4-bit BFP LLMs' performance with FP32 on downstream tasks.

## G  Searched mixed-precision LLMs

Mixed-precision quantisation is also helpful for recovering downstream task accuracy. Figure 7a and Figure 7b depict the performance of 4-bit LLMs on LAMBADA and ARC (easy) as the model scales up. The searched mixed-precision configuration effectively recovers the accuracy. Figure 8 and 9 is the activation distribution after searching on LAMBADA 2688 times. Keeping these layers in high precision effectively recovers the accuracy from 36.2% to 61.3% without decreasing the memory density, equivalent to a 4.3-bit OPT-2.7B.

## H  Mixed-precision with hardware model

We additionally performed a hardware-aware quantization search on BERT-Base (Devlin et al., 2018). We implemented the actual BFP hardware on FPGAs via high-level synthesis and model the hardware cost using Token per second (TPS) for speedup and TPS per LUT (TPS/LUT) for circuit area efficiency. Our hardware-aware search takes accuracy, memory bitwidth, and hardware cost as feedback. We compare the search traces of hardware-aware search and software-only search in Figure 10, and observe that the hardware-aware search curve is higher than the original one, proving our new objective function guides the search for better hardware efficiency. We will scale up experiments in future work.

| Method | Config | Model size | ARC (easy) | COPA | LAMBADA | PIQA | QNLI | SST2 | Average |
|---|---|---|---|---|---|---|---|---|---|
| FP32 | - | 125M | 43.5% | 66.0% | 37.9% | 63.0% | 49.4% | 53.3% | 52.7% |
| | | 350M | 44.0% | 72.0% | 45.2% | 64.4% | 49.5% | 61.8% | 57.5% |
| | | 1.3B | 57.0% | 79.0% | 57.9% | 71.7% | 51.3% | 82.2% | 69.6% |
| | | 2.7B | 60.8% | 77.0% | 63.6% | 73.9% | 51.1% | 51.7% | 65.4% |
| | | 6.7B | 65.6% | 81.0% | 67.7% | 76.3% | 50.9% | 76.5% | 73.4% |
| `LLM.int8()` | W8A8 | 125M | 43.6% | 66.0% | 37.7% | 63.0% | 49.5% | 52.1% | 52.5% |
| | | 350M | 43.8% | 72.0% | 45.3% | 64.2% | 49.5% | 66.1% | 58.3% |
| | | 1.3B | 57.5% | 79.0% | 57.7% | 71.6% | 51.1% | 80.1% | 69.2% |
| | | 2.7B | 60.6% | 78.0% | 62.9% | 72.9% | 51.2% | 52.1% | 65.3% |
| | | 6.7B | 65.5% | 83.0% | 66.6% | 76.1% | 50.7% | 76.4% | 73.5% |
| `LLM.int4()` | W8A8 | 125M | 41.5% | 65.0% | 34.3% | 62.1% | 49.5% | 51.1% | 50.8% |
| | | 350M | 41.6% | 68.0% | 44.6% | 64.0% | 49.5% | 60.8% | 55.8% |
| | | 1.3B | 55.9% | 78.0% | 54.5% | 70.0% | 51.7% | 76.6% | 67.0% |
| | | 2.7B | 58.7% | 77.0% | 62.2% | 73.7% | 51.3% | 50.8% | 64.5% |
| | | 6.7B | 64.5% | 81.0% | 66.2% | 74.7% | 51.6% | 76.4% | 72.5% |
| SmoothQuant-c | W8A8 | 125M | - | - | - | - | - | - | - |
| | | 350M | - | - | - | - | - | - | - |
| | | 1.3B | 55.8% | 78.0% | 55.3% | 71.2% | 51.1% | 75.9% | 67.2% |
| | | 2.7B | 60.4% | 77.0% | 64.0% | 72.6% | 51.3% | 51.7% | 65.2% |
| | | 6.7B | 65.3% | 81.0% | 68.5% | 74.7% | 51.1% | 71.6% | 72.2% |
| BFP | W6A6 | 125M | 42.6% | 69.0% | 37.1% | 62.6% | 49.4% | 51.6% | 52.6% |
| | | 350M | 43.7% | 72.0% | 42.8% | 65.1% | 49.6% | 64.6% | 57.6% |
| | | 1.3B | 57.0% | 79.0% | 51.8% | 71.6% | 51.5% | 79.7% | 67.8% |
| | | 2.7B | 60.9% | 76.0% | 64.1% | 73.3% | 49.9% | 53.1% | 65.5% |
| | | 6.7B | 65.2% | 80.0% | 67.2% | 75.8% | 50.9% | 76.1% | 72.9% |
| BFP | W4A4 | 125M | 37.0% | 65.0% | 28.7% | 58.9% | 49.1% | 49.5% | 47.8% |
| | | 350M | 39.9% | 65.0% | 38.7% | 58.9% | 49.3% | 55.9% | 51.7% |
| | | 1.3B | 50.0% | 71.0% | 41.4% | 65.7% | 50.2% | 58.1% | 57.2% |
| | | 2.7B | 52.4% | 70.0% | 36.2% | 68.0% | 51.4% | 51.7% | 55.7% |
| | | 6.7B | 61.5% | 84.0% | 56.0% | 73.1% | 52.3% | 61.1% | 67.2% |
| MiniFloat | W4A4 | 125M | 42.9% | 66.0% | 38.3% | 62.7% | 49.6% | 50.8% | 52.1% |
| | | 350M | 44.0% | 69.0% | 44.9% | 64.2% | 49.8% | 53.3% | 55.1% |
| | | 1.3B | 57.2% | 80.0% | 54.6% | 71.4% | 51.5% | 60.2% | 64.7% |
| | | 2.7B | 59.9% | 75.0% | 63.4% | 73.7% | 49.8% | 56.7% | 65.7% |
| | | 6.7B | 64.9% | 82.0% | 67.3% | 76.0% | 51.8% | 62.2% | 70.5% |

Table 7: A complete comparison of LLM quantisation methods on downstream tasks in the zero-shot prompting setup. QNLI, MRPC, and COLA results are not included because even FP32 LLMs yield an accuracy close to random prediction.

| Task | Fine-tuning style | Config | Model size | zero-shot prompting | epoch 0 | epoch 1 | epoch 2 |
|---|---|---|---|---|---|---|---|
| | | W32A32 | 125m | 53.3% | 91.2% | 92.9% | 92.6% |
| | FP32 | W32A32 | 350m | 61.8% | 92.3% | 93.1% | 93.4% |
| | | W32A32 | 1.3b | 82.2% | 93.9% | 93.2% | 94.0% |
| | | W32A32 | 2.7b | 51.7% | 94.6% | 94.5% | 94.7% |
| | | W5A5 | 125m | 49.5% (-3.8%) | 91.3% (0.1%) | 91.7% (-1.2%) | 92.0% (-0.6%) |
| SST2 | PTQ on downstream | W5A5 | 350m | 55.9% (-5.9%) | 92.7% (0.4%) | 92.5% (-0.6%) | 92.2% (-1.2%) |
| | | W5A5 | 1.3b | 58.1% (-24.1%) | 93.7% (-0.2%) | 93.2% (0.0%) | 93.6% (-0.4%) |
| | | W5A5 | 2.7b | 51.7% (0.0%) | 94.0% (-0.6%) | 95.3% (0.8%) | 94.5% (-0.2%) |
| | | W5A5 | 125m | 49.5% (-3.8%) | 92.0% (0.8%) | 92.0% (-0.9%) | 91.6% (-0.9%) |
| | TAQ on downstream | W5A5 | 350m | 55.9% (-6.0%) | 91.1% (-1.3%) | 91.6% (-1.5%) | 91.6% (-1.7%) |
| | | W5A5 | 1.3b | 58.1% (-24.1%) | 94.3% (0.3%) | 94.2% (0.9%) | 94.2% (0.1%) |
| | | W5A5 | 2.7b | 51.7% (0.0%) | 94.6% (0.0%) | 95.0% (0.4%) | 94.7% (0.0%) |
| | | W32A32 | 125m | 49.4% | 87.8% | 88.3% | 88.7% |
| | FP32 | W32A32 | 350m | 45.9% | 86.5% | 88.5% | 89.1% |
| | | W32A32 | 1.3b | 51.3% | 89.0% | 90.6% | 91.7% |
| | | W32A32 | 2.7b | 51.1% | 61.2% | 73.8% | 85.3% |
| | | W5A5 | 125m | 49.1% (-0.3%) | 82.0% (-5.8%) | 80.8% (-7.5%) | 85.7% (-2.0%) |
| QNLI | PTQ on downstream | W5A5 | 350m | 49.3% (-0.2%) | 85.5% (-0.9%) | 87.3% (-1.2%) | 88.2% (-0.9%) |
| | | W5A5 | 1.3b | 50.2% (-1.1%) | 88.6% (-0.5%) | 89.1% (-1.5%) | 90.3% (-1.4%) |
| | | W5A5 | 2.7b | 51.4% (0.2%) | 60.9% (-0.3%) | 71.6% (2.2%) | 84.2% (-1.1%) |
| | | W5A5 | 125m | 49.1% (-0.3%) | 86.1% (-1.7%) | 87.4% (-0.9%) | 88.2% (-0.5%) |
| | TAQ on downstream | W5A5 | 350m | 49.3% (-0.2%) | 85.5% (-1.0%) | 88.3% (-0.1%) | 88.6% (-0.5%) |
| | | W5A5 | 1.3b | 50.2% (-1.1%) | 86.6% (-2.5%) | 89.5% (-1.1%) | 91.1% (0.7%) |
| | | W5A5 | 2.7b | 51.4% (0.2%) | 86.2% (25.0%) | 88.1% (14.3%) | 92.5% (4.3%) |
| | | W32A32 | 125m | 0.0% | 41.2% | 47.3% | 49.8% |
| | FP32 | W32A32 | 350m | 0.0% | 13.0% | 39.8% | 47.2% |
| | | W32A32 | 1.3b | -6.9% | 29.1% | 56.3% | 56.9% |
| | | W32A32 | 2.7b | -3.5% | 0.0% | 8.0% | 25.9% |
| | | W5A5 | 125m | -1.1% (-1.1%) | 37.9% (-3.3%) | 40.4% (-6.9%) | 49.3% (-0.5%) |
| COLA[†] | PTQ on downstream | W5A5 | 350m | 0.0% (0.0%) | 7.5% (-5.5%) | 32.8% (-7.0%) | 46.0% (-1.2%) |
| | | W5A5 | 1.3b | -1.6% (5.3%) | 21.0% (-8.1%) | 51.9% (-4.4%) | 55.8% (-1.1%) |
| | | W5A5 | 2.7b | -3.1% (0.0%) | 0.0% (0.0%) | -2.1% (-10.1%) | 26.0% (0.1%) |
| | | W5A5 | 125m | -1.1% (-1.1%) | 43.1% (1.9%) | 41.1% (0.7%) | 43.7% (-5.6%) |
| | TAQ on downstream | W5A5 | 350m | 0.0% (0.0%) | 12.7% (-0.3%) | 31.5% (-1.3%) | 42.1% (-3.9%) |
| | | W5A5 | 1.3b | -1.6% (5.3%) | 33.9% (4.8%) | 49.0% (-2.9%) | 54.6% (-1.2%) |
| | | W5A5 | 2.7b | -3.1% (0.4%) | 0.7% (0.7%) | 0.0% (2.1%) | 18.6% (-7.4%) |
| | | W32A32 | 125m | 68.4% | 70.8% | 76.8% | 79.7% |
| | FP32 | W32A32 | 350m | 68.4% | 68.4% | 69.6% | 70.6% |
| | | W32A32 | 1.3b | 66.4% | 69.6% | 70.8% | 66.7% |
| | | W32A32 | 2.7b | 67.9% | 70.1% | 81.6% | 82.6% |
| | | W5A5 | 125m | 68.4% (0.0%) | 71.3% (0.5%) | 74.0% (-2.8%) | 78.8% (-0.9%) |
| MRPC | PTQ on downstream | W5A5 | 350m | 68.4% (0.0%) | 68.4% (0.0%) | 69.6% (0.0%) | 69.4% (-1.2%) |
| | | W5A5 | 1.3b | 57.8% (-8.6%) | 69.1% (-0.5%) | 70.1% (-0.7%) | 67.6% (0.9%) |
| | | W5A5 | 2.7b | 60.3% (-7.6%) | 68.6% (-1.5%) | 79.9% (-1.7%) | 78.9% (-3.7%) |
| | | W5A5 | 125m | 68.4% (0.0%) | 70.3% (-0.5%) | 71.6% (-5.2%) | 77.8% (-1.9%) |
| | TAQ on downstream | W5A5 | 350m | 68.4% (0.0%) | 69.4% (1.0%) | 69.7% (0.1%) | 70.6% (0.0%) |
| | | W5A5 | 1.3b | 57.8% (-8.6%) | 68.9% (-0.7%) | 71.1% (0.3%) | 71.1% (4.4%) |
| | | W5A5 | 2.7b | 60.3% (-7.6%) | 68.4% (-1.7%) | 68.4% (-13.2%) | 81.4% (-1.2%) |

Table 8: The comparison between PTQ fine-tuned FP32 and TAQ on SST2, QNLI, COLA, and MRPC. Both cases align 4-bit BFP LLMs with FP32 after fine-tuning. The latter may achieve slightly better accuracy. [†] means COLA is evaluated using the Matthews Correlation Coefficient (MCC), while the other tasks are evaluated using accuracy.

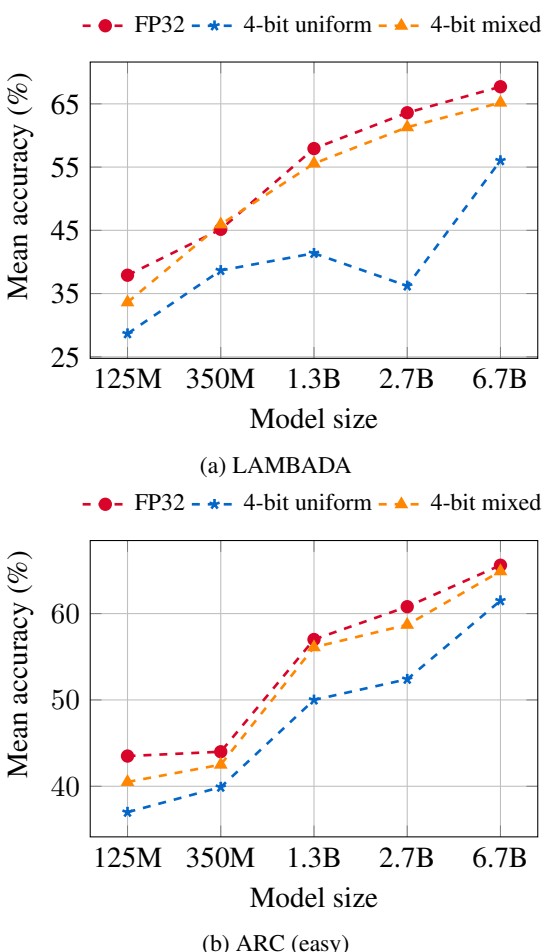

(a) LAMBADA

(b) ARC (easy)

Figure 7: The accuracy of FP32 model, 4-bit uniform BFP, and 4-bit mixed-precision on LAMBADA and ARCH (easy) in the zero-shot prompting context. Searched mixed-precision quantisation configuration captures the distribution inherent in LLMs, effectively recovering the accuracy.

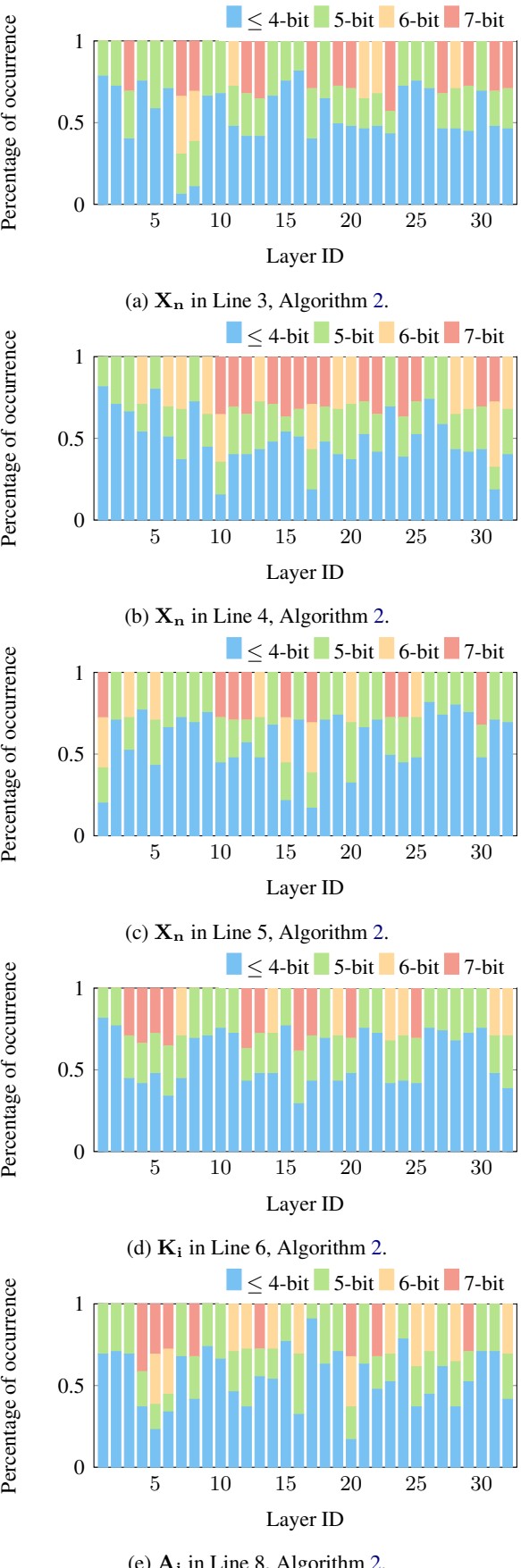

(a) $\mathbf{X_n}$ in Line 3, Algorithm 2.

(b) $\mathbf{X_n}$ in Line 4, Algorithm 2.

(c) $\mathbf{X_n}$ in Line 5, Algorithm 2.

(d) $\mathbf{K_i}$ in Line 6, Algorithm 2.

(e) $\mathbf{A_i}$ in Line 8, Algorithm 2.

Figure 8: The searched bit-width distribution of OPT-2.7B. Notably, some layers are frequently assigned relatively high precision, indicating these layers are less tolerant to quantisation.

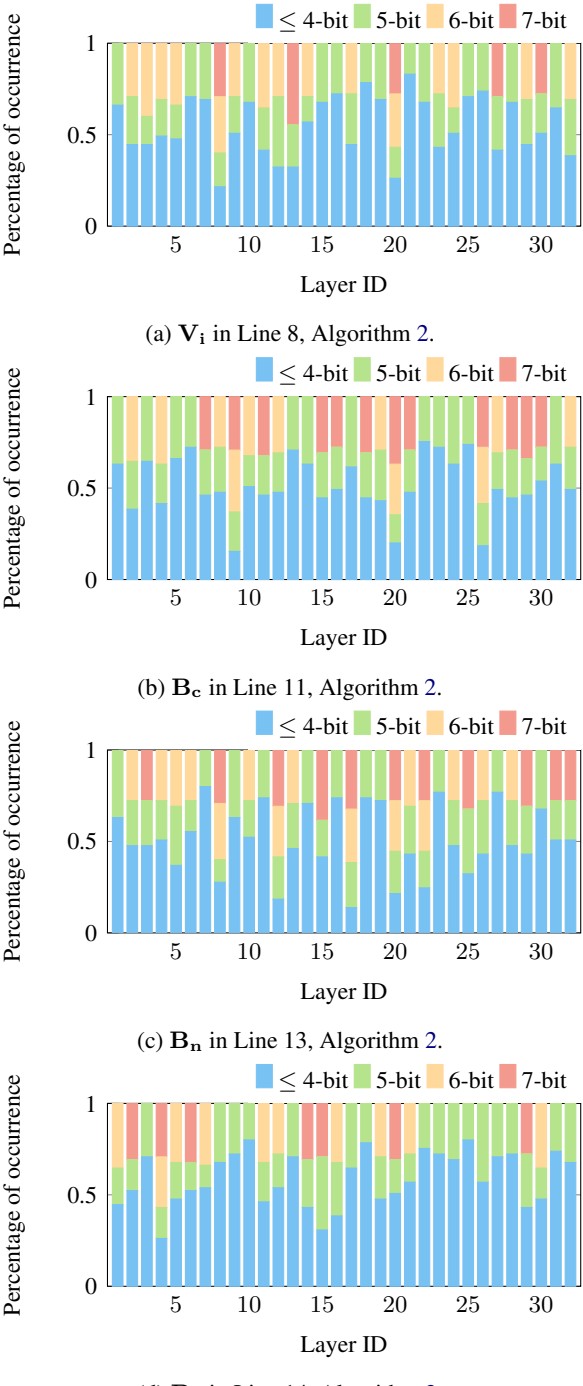

(a) $\mathbf{V_i}$ in Line 8, Algorithm 2.

(b) $\mathbf{B_c}$ in Line 11, Algorithm 2.

(c) $\mathbf{B_n}$ in Line 13, Algorithm 2.

(d) $\mathbf{B_1}$ in Line 14, Algorithm 2.

Figure 9: The searched bit-width distribution of OPT-2.7B. Notably, some layers are frequently assigned relatively high precision, indicating these layers are less tolerant to quantisation.

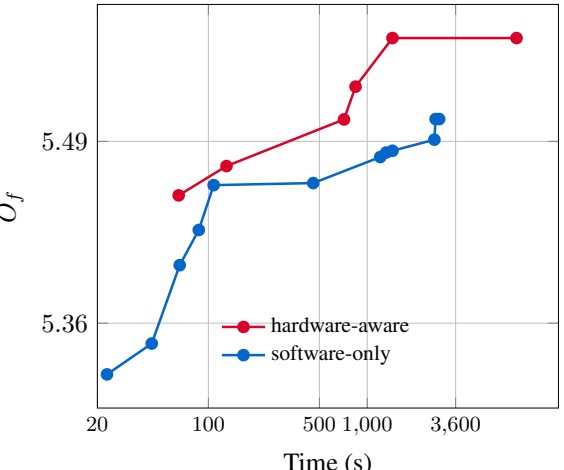

Figure 10: A comparison between the new hardware-aware search algorithm and the previous one that only considers accuracy and memory density. We introduce two additional hardware metrics, Token Per Second (TPS) and TPS Per LUT (TPS/LUT), such that the search algorithm can be aware of the hardware efficiency. The new objective function is $O_f = acc + \alpha_1 \cdot mem + \alpha_2 \cdot tps + \alpha_3 \cdot tpl$, where $tps$ and $tpl$ denote TPS and TPS/LUT respectively. We plot the traces of both searches versus search time for BERT-base. We observe that the hardware-aware curve is higher than the original one, proving our new objective function guides the search for better hardware efficiency in terms of speedup and circuit area.