# OpenReview forum: "Revisiting Block-based Quantisation: What is Important for Sub-8-bit LLM Inference?"
_EMNLP/2023/Conference — EMNLP 2023 Main_

### Official Review · Reviewer_F1oA · 2023-07-29

**Typos Grammar Style And Presentation Improvements:** Citations in Section 2.2 are broken.
**Soundness:** 4

**Excitement:**

4: Strong: This paper deepens the understanding of some phenomenon or lowers the barriers to an existing research direction.

**Paper Topic And Main Contributions:**

Authors address the problem of LLM quantization by exploring the combination of different floating point representations and precision levels. They find BFP, which shares an exponent across a block, performs better than various other alternatives (Fixed-point, MiniFloat, DMF, BM, BL). The proposed method is on par with LLM.int8 with 8 bits, a bit underperforms LLM.int4 with 4 bits, and the 6-bit version performs in between. Mixed precision further improves performance with minimal impact on memory density.

**Questions For The Authors:**

What exactly is the search space of hyperparameter optimization? In line 271-273, it seems like the granularity is a hyperparameter, and can chosen between layer-wise, tensor-wise, channel-wise, etc. Is this searched by HPO, or are they manually configured by authors? Does HPO algorithm choose the number of bits (4, 5, 6, 7, 8) for each of the block? That's my interpretation of "search for quantisation precision" (line 303) but I wasn't entirely sure, because the choice of precision across layers is only discussed in Mixed-precision experiments (Figure 3, line 499). Then, for the rest of experiments, do authors fix the number of bits, or do they always do the HPO search? Section 3.3 is written like HPO is always done.

Also, how does HPO iterations impact the performance of the model? Is it in general difficult to find a good configuration- do we consistently find a good configuration, or is there a high variance across runs?

When we use mixed precision, shouldn't authors be reporting how much memory density was sacrificed? In line 513 authors mention memory density was not compromised, but it'd help to provide a number (1%? 0.1%? 0.01%?)

**Reasons To Accept:**

Authors conduct a systematic study over block floating point representations. Observations of which representations work better than other representations provide good insight on the numerical nature of LLM inference. Therefore, future research on numerical aspects of LLMs will find these observations useful. Authors also estimate the impact of quantizations on ASIC accelerators, which might be useful for active research on building hardware accelerators. Caveat is that details of the estimation are deferred to Appendix.

**Reasons To Reject:**

I am not sure the proposed method has a comparative advantage over LLM.intX(). Table 3 doesn't have W8A8 version of BFP or LLM.int4(), so we cannot really do a head-to-head comparison with BFP with same bits. (Though LLM.int8 seems to only have 2x memory density, whereas other 8-bit precision methods in Table 3 have 4x-ish density; maybe I am terribly misunderstanding something.) In Table 4, 4-bit version of BFP W4A4 performs worse than LLM.int4(), and BFP W8A8 performs worse than LLM.int8(). 6-bit version BFP W6A6 is in between 4-bit and 8-bit counterparts, but if we are considering custom hardware chips, we could also consider LLM.int6() as a baseline?

I also found the description of hyperparameter search unclear, which makes it difficult to understand implications of experiments. I will leave them in questions for the authors.

**Reproducibility:**

4: Could mostly reproduce the results, but there may be some variation because of sample variance or minor variations in their interpretation of the protocol or method.

**Reviewer Confidence:**

3: Pretty sure, but there's a chance I missed something. Although I have a good feel for this area in general, I did not carefully check the paper's details, e.g., the math, experimental design, or novelty.

---

> ### Author Rebuttal · Authors · 2023-08-27
>
> > Q1: I am not sure the proposed method has a comparative advantage over LLM.intX(). Table 3 doesn't have W8A8 version of BFP or LLM.int4(), so we cannot really do a head-to-head comparison with BFP with same bits. (Though LLM.int8 seems to only have 2x memory density, whereas other 8-bit precision methods in Table 3 have 4x-ish density; maybe I am terribly misunderstanding something.) In Table 4, 4-bit version of BFP W4A4 performs worse than LLM.int4(), and BFP W8A8 performs worse than LLM.int8(). 6-bit version BFP W6A6 is in between 4-bit and 8-bit counterparts, but if we are considering custom hardware chips, we could also consider LLM.int6() as a baseline?
>
> Thanks for your questions. 8-bit/6-bit BFP outperforms LLM.int8() in terms of memory density and arithmetic density while maintaining comparable accuracy/perplexity to LLM.int8(). Here we answer the reviewer’s questions about LLM.int8()/int4() and provide a head-to-head comparison.
>
> - **Perplexity & accuracy**. We acknowledge that on OPT family from 125M to 6.7B, our 6-bit BFP’s performs slightly worse than LLM.int8() in terms of perplexity or accuracy. However, our 6-bit BFP has much higher hardware efficiency than LLM.int8() or LLM.int6() if it exists (See the detailed explanation in the next two paragraphs). Moreover, we further applied our experiments to newer and larger LLMs, and found our 6-bit BFP does not clearly perform worse than LLM.int8() in terms of perplexity. We assume this is because the numerical scaling offset described in Figure 1 is more obvious on larger models. Here are the perplexity results.
>
>   | Model           | FP32 |  LLM.int8()  | Ours 6-bit BFP |
>   |-----------------|:----:|:------------:|---------------:|
>   | LLAMA-7B        | 5.79 | 5.83 (+0.04) |   5.83 (+0.04) |
>   | Vicuna-7B       | 7.06 | **7.07 (+0.01)** |   7.08 (+0.02) |
>   | Alpaca-7B       | 7.01 | 7.02 (+0.01) |   7.02 (+0.01) |
>   | LLAMA-13B       | 5.17 | 5.22 (+0.05) |   **5.20 (+0.03)** |
>   | Vicuna-v1.5-13B | 6.13 | 6.16 (+0.03) |   6.16 (+0.03) |
>
>   Rebuttal Tab. 1: The perplexity ($\downarrow$) of a range of LLMs other than OPT quantized by our 6-bit BFP method. We compared our method with FP32 and LLM.int8() and found that our method achieves nearly lossless perplexity on Wikitext2.
>
>   In the table above, our 6-bit uniformed quantized BFP achieves almost the same perplexity as LLM.int8().
>
> - **Memory density**. LLM.int8()/LLM.int4() is different from plain 8-bit/4-bit fixed-point quantization. In LLM.int8()/LLM.int4() all the tensors are stored as 16-bit floating-points, which is the reason why both LLM.int8() and LLM.int4() have 2x memory density while plain 8-bit fixed-point has 4x memory density.
>
> - **Arithmetic density**. LLM.intX() targets GPUs while ours target AI accelerators. If LLM.int6() is mapped to a custom hardware chip, the chip’s arithmetic density **will be lower than 6-bit BFP**. LLM.intX() filters out outliers (the matrix elements with large magnitudes) using a manual threshold, casts inliers to X-bit, and performs an X-bit matrix multiplication for inliers and an FP16 matrix multiplication for outliers respectively. This thresholding is performed on the fly, which means the GPU or the custom hardware chip does not have prior knowledge about the sizes of the inlier matrix or the outlier matrix. The input-dependent FP16 matrix multiplication in LLM.intX() leads to a much lower arithmetic density than plain X-bit fixed-point or X-bit BFP on custom chips because the chip’s FP16 matrix multiply engine must fit the worst case to match the accuracy reported in LLM.intX() paper, i.e., the engine must accommodate the maximum possible number of FP16 outliers. The arithmetic density also depends on the accelerator architecture. For example, if the LLM.int6() matrix multiply engine is time-multiplexed, we need to consider the max possible number of outliers for all matrix multiplications this engine will process. In comparison, 6-bit uniform BFP does not require an FP16 matrix multiply engine. All tensors are stored and calculated in 6-bit.
>
> We will add the discussion of LLM.int8() to the revised version.
>
> ---
>
> > Q2: I also found the description of hyperparameter search unclear, which makes it difficult to understand implications of experiments. I will leave them in questions for the authors.
> What exactly is the search space of hyperparameter optimization? In line 271-273, it seems like the granularity is a hyperparameter, and can chosen between layer-wise, tensor-wise, channel-wise, etc. Is this searched by HPO, or are they manually configured by authors? Does HPO algorithm choose the number of bits (4, 5, 6, 7, 8) for each of the block? That's my interpretation of "search for quantisation precision" (line 303) but I wasn't entirely sure, because the choice of precision across layers is only discussed in Mixed-precision experiments (Figure 3, line 499). Then, for the rest of experiments, do authors fix the number of bits, or do they always do the HPO search? Section 3.3 is written like HPO is always done.
>
> **The granularity of quantization search is fixed to per-tensor in this work (the granularity is not a searchable hyper-parameter)**. For example, OPT-125m consists of 12 transformer blocks, each block has 8 matrix multiplications, and each matrix multiplication has two tensor operands (ignore the bias for now). If the bitwidth (i.e., precision) choices include (4,5,6,7,8), then the search space will be 5 ^ (2 \times 8 \times 12), i.e., each tensor can have a different precision. The linear layers with bias have one more hyper-parameter to search.
>
> **Only Section 4.4 presents the results of the mixed-precision search.** Sections 4.2 and 4.3 are both uniform quantizations without mixed precision. Here are the arguments we tried to claim in the paper,
>
> - Section 4.2: BFP enables uniform 6-bit post-training quantization on LLMs, outperforming existing methods in terms of accuracy and hardware efficiency (no mixed-precision search)
> - If we target sub-6-bit LLMs, we demonstrate two possible ways,
>   - Section 4.3: one is fine-tuning a uniform quantized LLM (no mixed-precision search)
>   - Section 4.4: the other is searching for a mixed-precision LLM in the context of post-training quantization.
>
>   We showed that both approaches allow 4-bit LLMs.
>
> ---
>
> > Q4: Also, how does HPO iterations impact the performance of the model? Is it in general difficult to find a good configuration- do we consistently find a good configuration, or is there a high variance across runs?
>
> We find more iterations usually give better mixed-precision results in terms of both accuracy and memory density. Besides the OPT family in the paper, we also applied the mixed-precision search on BERT-large, LLAMA-7B, Alpaca-7B, and Vicuna-7B. We find that 128 iterations (n_trials=128) are enough for TPE to find promising results on all the models above.
>
> In terms of the variance across runs, we calculated the mean and variance of 42 runs (each run has 128 iterations) of OPT-2.7B on LAMBADA (we used these runs to visualize the bitwidth distribution in Figure 3). Here is the variance and mean of accuracy and memory density across runs,
>
> |          | 0-shot Accuracy$^*$ ($\uparrow$) | Memory density ($\uparrow$) | Average bitwidth ($\downarrow$) |
> |----------|:------------:|:--------------:|-----------------:|
> | Mean     |     0.6252    |      7.41$\times$     |             4.31 |
> | Variance |   2.81E-04   |    4.48E-05    |         1.52E-05 |
>
> Rebuttal Tab. 5: The variance of our search runs of OPT-2.7b on LAMBADA. We calculated the variance and mean of accuracy and memory density of 42 runs where each run consisted of 128 trials and found that the run variance is small, proving that our search method consistently finds promising mixed-precision configurations. $^*$ means the accuracy is reported on the training set of SST2, which is reasonable since we are performing post-training quantization without fine-tuning the LLM on the training set.
>
> We observe that the variance of the 42 runs is small, which means the searched mixed-precision networks are consistently promising.
>
> ---
>
> > Q5: When we use mixed precision, shouldn't authors be reporting how much memory density was sacrificed? In line 513 authors mention memory density was not compromised, but it'd help to provide a number (1%? 0.1%? 0.01%?)
>
> Thanks for your questions and apologize for our unclearness. The memory density of the search OPT-2.7B in line 513 is 7.42x, which is even better than the uniform 4-bit BFP’s 7.11x. We will clarify this value in the revised version.
>
> In this paper, the memory density is equivalent to the average bitwidth. We follow [this work](https://proceedings.neurips.cc/paper/2020/hash/747e32ab0fea7fbd2ad9ec03daa3f840-Abstract.html) to calculate the memory density by normalizing the average bitwidth to FP32.
> $$
> mem = \frac{32}{aw}
> $$
> where $mem$ and $aw$ denote memory density and average bitwidth respectively.

---

### Official Review · Reviewer_ra67 · 2023-08-03

**Typos Grammar Style And Presentation Improvements:** 1. In line 4 in the Abstract, it shou…
**Soundness:** 4

**Excitement:**

4: Strong: This paper deepens the understanding of some phenomenon or lowers the barriers to an existing research direction.

**Paper Topic And Main Contributions:**

The paper explores the problem of quantisation in Large Language Models (LLMs), which is used to reduce the computational and memory resources required for running inference on these models by representing the model parameters with a lower precision format. However, current quantisation techniques like LLM.int8() and SmoothQuant focus on 8-bit precision, and a further reduction in bits causes large accuracy drops. The authors attribute numerical scaling offsets as the main bottleneck to LLM quantisation. To address this problem, they propose adapting block-based quantisation methods for LLMs like Block Floating Point (BFP). These methods share scaling factors across packed numbers, thereby efficiently reducing numerical scaling offsets without necessitating additional treatments in the computational path.

The authors demonstrate that their block-based quantisation framework can achieve nearly lossless quantised 6-bit LLMs. These surpass prior work in terms of arithmetic and memory density without requiring any data calibration or re-training. Additionally, the paper provides insights into sub-8-bit LLM quantisation, including the impact of mismatches between activation and weight distributions, optimal fine-tuning strategies, and lower quantisation granularity inherent in the statistical properties of LLMs. The authors show that the latter two methods enable nearly lossless 4-bit LLMs on downstream tasks. Mixed-precision search further demonstrates the potential advantage of shifting LLM inference to cost-effective ASICs. The authors conduct experiments to validate their approach, comparing their block-based quantisation technique with traditional methods on several language processing tasks. Their method outperforms conventional methods in terms of perplexity, accuracy, memory density, and arithmetic density.

**Questions For The Authors:**

The authors should address the weaknesses mentioned in the above section.

**Reasons To Accept:**

1. The paper presents a new perspective on the problem of LLM quantisation by focusing on the issue of numerical scaling offsets.

2. The authors adapt block-based quantisation methods for LLMs like Block Floating Point (BFP), demonstrating that these methods can efficiently reduce numerical scaling offsets without requiring additional treatments in the computational path.

3. The paper shows nearly lossless 6-bit quantized LLMs using BFP, which achieve 19x higher arithmetic density and 5x higher memory density compared to the float32 baseline. This 6-bit BFP outperforms prior work like LLM.int8() and SmoothQuant by 2.5x in arithmetic density and 1.2x in memory density without requiring data calibration or re-training.

3. The paper offers valuable insights into sub-8-bit LLM quantisation, including the impact of mismatches between activation and weight distributions, optimal fine-tuning strategies, and the implications of the lower quantisation granularity inherent in the statistical properties of LLMs.

4. The authors present two methods to achieve nearly-lossless 4-bit quantisation on downstream tasks, including fine-tuning-based and mixed-precision search.

5. The paper also explores that mixed-precision search has the potential advantage of shifting LLM inference to cost-effective ASICs.

6. The authors conduct extensive experiments to validate their proposed approach, comparing their block-based quantisation technique with traditional methods on several language processing tasks. Their method consistently outperforms traditional methods across perplexity, accuracy, memory density, and arithmetic density.

7. The paper is well-written and well-structured.

**Reasons To Reject:**

1. The authors highlight that they have not implemented the quantisation methods on GPU systems to demonstrate real speedups due to a lack of CUDA kernel implementation.

2. The paper also mentions that the search algorithm does not include arithmetic density due to a lack of hardware models.

3. Although the authors have mentioned the limitations in the paper, they should provide a more detailed plan on how they plan to address these drawbacks in their future work.

4. The evaluation is limited to language modeling and downstream NLP tasks. Testing the proposed quantisation methods on other modalities like computer vision could benefit the paper.

5. The authors should provide detailed ablation studies to isolate the impact of block-based quantisation.

**Reproducibility:**

4: Could mostly reproduce the results, but there may be some variation because of sample variance or minor variations in their interpretation of the protocol or method.

**Reviewer Confidence:**

3: Pretty sure, but there's a chance I missed something. Although I have a good feel for this area in general, I did not carefully check the paper's details, e.g., the math, experimental design, or novelty.

---

> ### Author Rebuttal · Authors · 2023-08-27
>
> We highly appreciate the reviewer's helpful suggestions and questions. Here we answer the questions one by one.
>
> > Q1: The authors highlight that they have not implemented the quantisation methods on GPU systems to demonstrate real speedups due to a lack of CUDA kernel implementation.
>
> In this paper, we try to highlight that block-based quantization arithmetic has the potential to be the next-generation AI accelerator for LLM inference (domain-specific hardware) rather than GPGPUs. After we submitted the draft, we finished a series of experiments of mixed-precision BFP LLMs on FPGA-based AI accelerators where BFP computation units were implemented. Our FPGA-based accelerator outperforms GPUs in terms of hardware efficiency without sacrificing accuracy. Here are our results,
>
> |                       |   Models  | 0-shot Accuracy (%) | Mean bitwidth | SPS ($\uparrow$) | SPS/W ($\uparrow$) |
> |-----------------------|:---------:|:-------------------:|:-------------:|:----------------:|-------------------:|
> | GPU (RTX 3090)        |  OPT-2.7B |        51.27        |       32      |        69        |          **0.051** |
> |                       |  OPT-6.7B |        76.49        |       32      |         6        |              0.008 |
> |                       | Alpaca-7B |        84.98        |       32      |         7        |              0.008 |
> | Our mixed-BFP on FPGA |  OPT-2.7B |        62.61        |      4.0      |      **85**      |              0.038 |
> |                       |  OPT-6.7B |        72.94        |      4.1      |      **33**      |          **0.012** |
> |                       | Alpaca-7B |        89.06        |      4.5      |      **25**      |          **0.011** |
>
> Rebuttal Tab. 4: a comparison between FP32 LLMs on GPUs and mixed-precision BFP LLMs on FPGAs. We report zero-shot prompting accuracy on SST2. The last two columns are hardware metrics. We use Sequence Per Second (SPS) and SPS per Watt (SPS/W) to evaluate hardware efficiency. Sequence per second is the number of text sequences that our FPGA-based accelerator processes per second, which estimates the inference throughput. SPS per Watt measures the SPS our accelerator reaches per unit of power, which estimates the energy efficiency. Our FPGA-based accelerator reaches SPS and SPS/W higher than GPUs.
>
> We use Sequence Per Second (SPS) and SPS per Watt (SPS/W) to evaluate hardware efficiency and report zero-shot prompting accuracy. Sequence per second is the number of text sequences that our FPGA-based accelerator processes per second, which estimates the inference throughput. SPS per Watt measures the SPS that our accelerator reaches per unit of power, which estimates the energy efficiency. Our SPS and SPS/W are usually higher than GPUs, proving that the mixed-BFP actually speeds up the LLM inference and improves power efficiency compared to GPUs.
>
> ---
>
> > Q2: The paper also mentions that the search algorithm does not include arithmetic density due to a lack of hardware models.
>
> We acknowledge that the lack of arithmetic density is our limitation. To solve this problem, we plan to introduce more accurate hardware metrics (Sequence Per Second (SPS) and SPS per LUT (SPS/LUT)) into our search algorithm. Please refer to the answer to Q3 below for a detailed explanation.
>
> ---
>
> > Q3: Although the authors have mentioned the limitations in the paper, they should provide a more detailed plan on how they plan to address these drawbacks in their future work.
>
> We will provide a more detailed plan for future work in the revised version. We plan to introduce two more accurate hardware metrics than arithmetic density for modeling the hardware cost.
>
> Basically, we can implement the actual BFP hardware on FPGAs via high-level synthesis and model the hardware cost (SPS for speedup and SPS/LUT for circuit area efficiency) using polynomial regression. In this way, our search algorithm can be aware of the hardware cost of quantization configurations. The improved search objective will become
> $$
> Q_f = acc + \alpha_1 \cdot mem + \alpha_2 \cdot sps + \alpha_3 spl
> $$
> where $sps$ and $spl$ denote SPS and SPS/LUT.
>
> We have verified the feasibility of this method on BERT-base and compared the new hardware-aware search algorithm with the previous one.
>
> [Rebuttal Fig2: Comparison between the search with and without SPS and SPS/LUT](https://i.imgur.com/Tl2dtYX.png)
>
> By comparing the traces of the two search methods (with/without SPS and SPS/LUT), we observe that introducing hardware metrics indeed guides the search to find more hardware-efficient mixed-precision models. We leave scaling up experiments on LLMs (such as LLAMA) as future work.
>
> ---
>
> > Q4: The evaluation is limited to language modeling and downstream NLP tasks. Testing the proposed quantization methods on other modalities like computer vision could benefit the paper.
>
> We mainly target LLM quantization when drafting this paper as the observed increasing variance in LLMs (Figure 1) motivates our research. In our humble opinion, the quantization of vision models is out of scope of this paper but maybe a promising future direction.
>
> ---
>
> > Q5: The authors should provide detailed ablation studies to isolate the impact of block-based quantization.
>
> Thanks for your suggestions. We did not explicitly include an ablation study section, but the impact of block-based quantization can be observed in Table 3 by comparing block-based quantization with fixed-point quantization and MiniFloat.
>
> 8-bit fixed-point quantization can be considered as an extreme case of 8-bit BFP where the block size equals the tensor size. MiniFloat or standard floating-point is another extreme case of BFP where the block size is 1. We can observe that the accuracy of quantized LLMs increases as the block size decreases (fixed-point -> BFP -> MiniFloat) but the hardware efficiency decreases as well.
>
> We will add this analysis to the revised version.

---

### Official Review · Reviewer_Qm4i · 2023-08-04

**Soundness:** 3

**Excitement:**

4: Strong: This paper deepens the understanding of some phenomenon or lowers the barriers to an existing research direction.

**Paper Topic And Main Contributions:**

This paper explores non-linear quantization methods applied to LLM inference (up to 6.7B). It provides insights in LLM quantization and analyzes arithmetic and memory density of various quantization methods.

Instead of scaling blocks by normalizing to a predefined interval which introduces quantization constants, this paper suggests a combination of approaches:
1. non-linear quantization using Block Mini-Float, Block Floating Point, and Block Logarithm methods. These methods share an exponent or exponent bias for elements of a block.
2. implement these methods on FPGAs.
3. use quantization search to optimize the quantization configuration for each layer of the transformer.

**Reasons To Accept:**

- Insights in quantization methods which lead to improved arithmetic and memory density with little perplexity degradation.
- Interesting demonstration of use of FPGAs to improve the efficiency of LLM inference though quantization.

**Reasons To Reject:**

Because of the quantity of information presented, the paper comes across as difficult to follow and scattered and the value of the insights is greatly diminished. Some of the elements that would benefit more clarification and further ablations to justify the design choices:
- What are the tradeoffs of using FPGAs and GPUs?
- Ablations with and without quantization search. How far can you go with the non-linear quantization methods BM, BFP, BL without quantization search? How important is it? Can it be used with other quantization methods?
- The finetuning discussion appears to be out of place in the paper and it does not add to the general argument of the paper.

**Reproducibility:**

3: Could reproduce the results with some difficulty. The settings of parameters are underspecified or subjectively determined; the training/evaluation data are not widely available.

**Reviewer Confidence:**

3: Pretty sure, but there's a chance I missed something. Although I have a good feel for this area in general, I did not carefully check the paper's details, e.g., the math, experimental design, or novelty.

---

> ### Author Rebuttal · Authors · 2023-08-27
>
> > Q1: What are the tradeoffs of using FPGAs and GPUs?
>
> Thank you for your questions. In order to provide a comprehensive comparison of the tradeoffs, it is better if we look at between GPU, ASIC (Application-Specific Integrated Circuits), and FPGAs (Field Programmable Gate Arrays) for accelerating custom non-linear arithmetics.
>
> It is worth noting that there is a growing trend towards utilizing domain-specific hardware for accelerating ML workloads, exemplified by Google's TPU (v1-v5) for AI acceleration (ASIC accelerator), Microsoft BrainWave project (FPGAs) as compared to NVIDIA's GPGPU. In this regard, we hold the belief that custom hardware optimized for efficiency and cost will be the future generation hardware for LLM inference. FPGAs, however, serve as an intermediary between GPUs and ASICs. Due to their arithmetic density and memory density, we consider FPGAs to be a viable approximation of ASICs in our analysis.
>
> NVIDIA’s GPU is the most common choice for training and testing neural networks in the deep learning community because deep learning frameworks like PyTorch have high-performance tensor computation libraries built on CUDA e.g., Aten for PyTorch. However, when it comes to deployment/inference for production (no training is performed), ASICs for neural networks, also known as AI accelerators, are a better choice than GPUs in terms of efficiency (speed, energy, etc.). Another limitation of GPUs is the lack of native support for custom quantization arithmetics like the int4, BFP, BL, and BM mentioned in this paper. For example, NVIDIA RTX 4090 only supports FP64, FP32, Bfloat16, FP16, and Int8, and PyTorch supports training & testing with FP64, FP32, Bfloat16, and FP16. Though sometimes it is possible to support custom quantization arithmetics via manually-designed CUDA kernel, the efficiency is still much lower than ASICs.
>
> FPGAs are reconfigurable hardware that can be programmed to meet the developer’s requirements. The programmability of FPGAs makes it a good choice for implementing computation units for custom quantization arithmetics. There have been considerable publications on accelerating neural networks on FPGAs. Another advantage of FPGAs is that the development life circle is much shorter than ASICs, thus FPGAs are suitable for research. Additionally, FPGA implementations are typically perceived as a valuable testbed for ASIC designs. The higher performance numbers achieved through FPGAs directly correlate with the potential performance of corresponding ASIC designs, which can take advantage of the faster clock speed.
>
> In conclusion, GPUs are a good choice for training but FPGAs have higher efficiency in terms of accelerating the inference of custom non-linear quantization arithmetics like BFP.  Moreover, the programmability of FPGAs enables us to report the arithmetic density of the custom BFP computation units via logic synthesis. It will be hard to model the memory density and arithmetic density on GPUs even if we have CUDA kernels implemented.
>
> ---
>
> > Q2: Ablations with and without quantization search. How far can you go with the non-linear quantization methods BM, BFP, BL without quantization search? How important is it? Can it be used with other quantization methods?
>
> We have included a series of ablation studies in the paper but maybe didn’t clearly point out where they are. We will clarify our observation (with/without search) in the revised version. Here we split your question into two parts and address them one by one.
>
> > > Q2.1: Ablations with and without quantization search. How far can you go with the non-linear quantization methods BM, BFP, BL without quantization search? How important is it?
>
> Section 4.2 compares uniform quantization arithmetics (without a quantization search), and Section 4.3 is the only section that performs a quantization search. Here are the ablation studies we conducted.
>
> - For BFP, 6-bit is the minimum bitwidth we could achieve while maintaining comparable accuracy/perplexity to baselines.
>   - Line 510 is our observation that “By preserving high precision for these sensitive layers, we recovered the 4-bit LLM accuracy from 36.2% to 61.3% on LAMBADA without compromising memory density”. Here the zero-shot prompting accuracy of 36.2% is the one without quantization search and 61.3% is the one using quantization search. Figure 8 in the appendix is a more comprehensive ablation study of mixed-precision search including a range of model sizes on the two datasets. In Figure 8 the curve “4-bit uniform” is 4-bit BFP quantization without search and the curve “4-bit mixed” means that the searched mixed-precision LLM has an average bitwidth of 4. This figure implies that quantization search is an effective way to recover the accuracy for 4-bit LLMs.
>
>     [shortcut to Figure 8 in the paper](https://imgur.com/ygKtncK)
>
>   - Table 3 shows the uniform 6-bit BFP and 4-bit BFP without quantization search on wikitext2. We also have results of 5-bit BFP but we did not include it in Table 3 due to the page limit. We instead drew a conclusion that “We also observe that sub-6-bit BFP has a severe accuracy drop” in line 442.
>
>     [Shortcut to Table 3 in the paper](https://i.imgur.com/ERrgibE.png)
>   - Here is a table of the perplexity comparison of BFP bitwidth. We can see that 5-bit and 4-bit BFP have increased perplexity with an average of around 1 and 10 respectively compared to FP32, which is worse than existing state-of-the-art quantization schemes like LLM.int8() and SmoothQuant. We will add the “BFP W5A5” row if the paper is accepted since the camera-ready version allows one additional page.
>
>     |          |      OPT-125M      |      OPT-350M      |      OPT-1.3B      |      OPT-2.7B     |   OPT-6.7B |
>     |----------|:--------------:|:--------------:|:--------------:|:-------------:|--------------:|
>     | FP32     |      27.65     |      22.00     |      14.62     |     12.47     |         10.86 |
>     | BFP W6A6 |  28.27 (+0.62) |  22.22 (+0.22) |  15.08 (+0.46) | 12.54 (+0.07) | 10.90 (+0.04) |
>     | BFP W5A5 |  29.91 (+2.26) |  23.77 (+1.77) |  15.96 (+1.34) | 13.01 (+0.54) | 11.23 (+0.37) |
>     | BFP W4A4 | 41.94 (+14.29) | 33.98 (+11.98) | 24.70 (+10.07) | 19.34 (+6.87) | 13.59 (+2.73) |
>
>     Rebuttal Tab. 2: A perplexity ($\downarrow$) comparison of 4-bit, 5-bit, and 6-bit BFP on Wikitext2. We observe that 6-bit BFP is the lowest bitwidth we can achieve without mixed-precision quantization search or fine-tuning.
>   - Table 4 shows the uniform 6-bit, 5-bit, and 4-bit BFP without quantization search on five downstream tasks (Table 4 reports mean zeros-shot prompting accuracy, and Table 6 in the appendix reports complete results). In Table 4, we also observe that 6-bit BFP is the best we can do without compromising accuracy. 5-bit BFP OPT-1.3B has a mean accuracy drop of 4.1%.
>
>     [shortcut to Table 4 in the paper](https://i.imgur.com/AUt5DJc.png)
>
> - For BM and BL, as we observed in Table 3, even 8-bit BL or 8-bit BM has a perplexity that is much higher than existing baselines such as LLM.int8() and SmoothQuant (8-bit BL and BM’s perplexity is increased by ~20x and ~800x respectively), mixed-precision search for 8-bit BL or BM will not find promising results. We drew this conclusion in line 430 of the paper and excluded BL and BM from the mixed-precision search and ablation study.
>
> >> Q2.2: Can it be used with other quantization methods?
>
> Yes, basically the search objective in line 312 can be extended to support other quantization methods if software modeling or hardware modeling is provided. Here we provide an example of extending our current search method.
>
> Example: mixed-precision fixed-point quantization search.
>
> We add the memory density model of fixed-point quantization to our search algorithm, and search for mixed-precision fixed-point quantized LLMs (LLAMA-7B, Vicuan-7B, Alpaca-7B, and OPT-6.7B) on SST2. Here are our results,
>
> | Models    | Fixed-point W8A8 | *Mixed-fixed-point* |   BFP W8A8  |   Mixed-BFP   |         FP32 |
> |-----------|:----------------:|:-----------------:|:-----------:|:-------------:|-------------:|
> | LLAMA-7B  |    (53.33, 8)   |  *(62.50, 4.7)* | (83.14, 8) | (84.38, 4.5) | (83.26, 32) |
> | Vicuna-7B |    (53.33, 8)   | *(58.59, 5.1)* | (8028, 8) | (82.91, 4.4) | (83.94, 32) |
> | Alpaca-7B |    (55.85, 8)   | *(58.59, 5.1)* | (81.65, 8) | (89.06, 4.5) | (84.98, 32) |
> | OPT-6.7B  |    (49.08, 8)   | *(53.91, 4.7)* | (76.15, 8) | (72.94, 4.4) | (76.49, 32) |
>
>  Rebuttal Tab. 3: The (zero-shot prompting accuracy (%), average bitwidth) of mixed-precision fixed-point quantization search. We extended our search method and searched for mixed-precision strategies of fixed-point quantized LLMs on SST2. We observed that the searched fixed-point LLMs achieved higher accuracy than uniform 8-bit fixed-point, but had lower average bitwidth.
>
> ---
>
> > Q3: The finetuning discussion appears to be out of place in the paper and it does not add to the general argument of the paper.
>
> Sorry for any confusion. Respectfully, we believe that finetuning plays a significant role in our contribution. In this paper, we present the following arguments:
> - BFP enables uniform 6-bit post-training quantization on LLMs, and significantly outperform existing methods in terms of accuracy and hardware efficiency.
> - If we target sub-6-bit LLMs, there are two possible ways,
>   - one is fine-tuning a uniform quantized LLM (no mixed-precision search), which is also commonly known as Quantization Aware Training (QAT).
>   - the other is searching for a mixed-precision LLM in the context of post-training quantization (no fine-tuning).
>
>   We demonstrated both approaches allow 4-bit LLMs.
>
> We know LLMs pre-trained on language modeling tasks may fail on some downstream tasks such as QNLI and COLA, thus fine-tuning is necessary. This then stems the question: if we apply quantization, should we (1) quantize the pre-trained LLM first, then do quantization-aware training on downstream tasks, or (2)  fine-tune the pretrained FP32 LLM first, then do post-training-quantization? We compare both options on a range of downstream tasks that zero-shot prompting cannot handle (fine-tuning is necessary even for FP32 pre-trained LLMs) in the fine-tuning subsection (Section 4.3) and drew the following conclusion in line 469,
>
> - Both options can achieve nearly lossless 4-bit LLMs in terms of accuracy on challenging downstream tasks.
> The fine-tuning of (1) is easier to converge due to the Straight-Through Estimator (STE) in (2)’s backward pass.
>
> - Moreover, prior work like LLM.int8(), SmoothQuant, and GPTQ only consider PTQ of LLM, but our work **considers both PTQ and QAT (PTQ for uniform 6-bit and 4-bit mixed-precision search, QAT for fine-tuning 4-bit)**.

---

### Official Review · Reviewer_JYyW · 2023-08-06

**Soundness:** 4

**Excitement:**

4: Strong: This paper deepens the understanding of some phenomenon or lowers the barriers to an existing research direction.

**Paper Topic And Main Contributions:**

This paper investigates quantization methods for large language models. Through analysis of activations, the authors indicate that variances tend to increase in deeper layers of the network, yielding scaling offsets in their quantization. The work studies a variety of block arithmetic based methods. The paper contributes a wealth of empirical analysis encompassing multiple tasks / settings as well as model sizes.

**Questions For The Authors:**

* How should we think about the results applying to models other than OPT? Would they likely hold? What would be some of the factors that could enable us to see the same/different results?


**Reasons To Accept:**

The paper presents a seemingly comprehensive study of quantization methods for LLMs with a focus on 8-bits or less. These results include a comparison to both:
* baseline methods such as LLM.int8(), SmoothQuant, GPTQ
* block based methods: BM, BFP, BL, etc.

The results compare performance on a variety of task settings: WikiText2, LAMBADA, QNLI, lm-eval-harness, and many others. The proposed approaches in this paper achieve some of the best results, e.g., Table 3. This includes the fine-tuning setting (both PTQ/TAQ).

The paper presents the approaches in a thoughtful way with experiments that provide a detailed landscape of quantization in LLMs. This could be a benefit to readers understanding the design space and research landscape.

**Reasons To Reject:**

Concerns about the paper include:
* The authors select OPT to have a variety of model sizes across their experiments. If I have understood correctly the OPT models are the only models used in experiments. I wonder how much we would expect the results to hold across other pretrained models?

**Reproducibility:**

3: Could reproduce the results with some difficulty. The settings of parameters are underspecified or subjectively determined; the training/evaluation data are not widely available.

**Reviewer Confidence:**

3: Pretty sure, but there's a chance I missed something. Although I have a good feel for this area in general, I did not carefully check the paper's details, e.g., the math, experimental design, or novelty.

**Typos Grammar Style And Presentation Improvements:**

The highlighting in Table 4 might be more effective by defining what close means and then highlighting all methods that meet that definition.

---

> ### Author Rebuttal · Authors · 2023-08-27
>
> We thank the reviewers for the valuable feedback.
>
> > Q1: The authors select OPT to have a variety of model sizes across their experiments. If I have understood correctly the OPT models are the only models used in experiments. I wonder how much we would expect the results to hold across other pretrained models?
>
> Following your suggestion, we extended our methods to other LLMs. The new results on other LLMs further support the claims in our paper.
>
> The table below shows the comparison of our 6-bit uniformly quantized BFP LLMs with the baselines. We now include LLAMA, Vicuna, and Alpaca. We observe that 6-bit BFP archives better hardware efficiency under the same perplexity constraint. We will add these new results to Table 2 in the revised version.
>
> | Model           | FP32 |  LLM.int8()  | Ours 6-bit BFP |
> |-----------------|:----:|:------------:|---------------:|
> | LLAMA-7B        | 5.79 | 5.83 (+0.04) |   5.83 (+0.04) |
> | Vicuna-7B       | 7.06 | **7.07 (+0.01)** |   7.08 (+0.02) |
> | Alpaca-7B       | 7.01 | 7.02 (+0.01) |   7.02 (+0.01) |
> | LLAMA-13B       | 5.17 | 5.22 (+0.05) |   **5.20 (+0.03)** |
> | Vicuna-v1.5-13B | 6.13 | 6.16 (+0.03) |   6.16 (+0.03) |
>
> Rebuttal Tab. 1: The perplexity ($\downarrow$) of a range of LLMs other than OPT quantized by our 6-bit BFP method. We compared our method with FP32 and LLM.int8() and found that our method achieves nearly lossless perplexity on Wikitext2.
>
> ---
>
> > Q2: How should we think about the results applying to models other than OPT? Would they likely hold? What would be some of the factors that could enable us to see the same/different results?
>
> Our answer is that our conclusion still holds for other LLMs. The intuitive explanation is that all these models share similar network architectures (Transformer) and are trained on language modeling tasks. All of these LLMs would have the numerical scaling offsets issue described in the introduction section.
>
> To verify our hypothesis, we performed two groups of experiments.
> - First, we plotted the tensor variances of LLMs other than OPT, and observed the same increasing trends as Figure 1 in the paper. Similar to OPT-6.7B, Vicuna-7B’s activation variance also increases with the layer number and is consistently larger than the weight variance.
>
>   [Rebuttal Fig.1: variance curves of Vicuna-7B](https://imgur.com/LOo5m6X)
>
> - Second, we quantized newer and larger LLMs out of OPT family and compared their perplexity with baselines. Here is the result table (the same one in the answer of Q1).
>
>   | Model           | FP32 |  LLM.int8()  | Ours 6-bit BFP |
>   |-----------------|:----:|:------------:|---------------:|
>   | LLAMA-7B        | 5.79 | 5.83 (+0.04) |   5.83 (+0.04) |
>   | Vicuna-7B       | 7.06 | **7.07 (+0.01)** |   7.08 (+0.02) |
>   | Alpaca-7B       | 7.01 | 7.02 (+0.01) |   7.02 (+0.01) |
>   | LLAMA-13B       | 5.17 | 5.22 (+0.05) |   **5.20 (+0.03)** |
>   | Vicuna-v1.5-13B | 6.13 | 6.16 (+0.03) |   6.16 (+0.03) |
>
>   Rebuttal Tab. 1: The perplexity ($\downarrow$) of a range of LLMs other than OPT quantized by our 6-bit BFP method. We compared our method with FP32 and LLM.int8() and found that our method achieves nearly lossless perplexity on Wikitext2.
>
> We will add the results of other LLMs to the appendix of the revised version.

---

### Meta-Review · Area_Chair_6QAy · 2023-09-19

**Recommendation:** 5

**Metareview:**

The paper presents a thorough investigation into quantization methods for Large Language Models (LLMs), focusing on 8-bits or less. Reviewers commended the comprehensive comparison against both baseline and block-based methods across multiple task settings, emphasizing the innovative introduction of non-linear quantization strategies. Particularly notable are the impressive results achieved with nearly lossless 6-bit and 4-bit quantized LLMs using Block Floating Point (BFP), surpassing existing methods in arithmetic and memory density. The study's detailed insights, combined with its well-structured presentation, make it a significant contribution to both numerical LLM research and potential applications in hardware accelerators.

Pros:

extensive experiments and well written

Cons:

Could use more explanation on some ablations

---

### Decision · Program_Chairs · 2023-10-07

**Decision:**

Accept-Main

**Comment:**

The paper presents a thorough investigation into quantization methods for Large Language Models (LLMs), focusing on 8-bits or less. Reviewers commended the comprehensive comparison against both baseline and block-based methods across multiple task settings, emphasizing the innovative introduction of non-linear quantization strategies. Particularly notable are the impressive results achieved with nearly lossless 6-bit and 4-bit quantized LLMs using Block Floating Point (BFP), surpassing existing methods in arithmetic and memory density. The study's detailed insights, combined with its well-structured presentation, make it a significant contribution to both numerical LLM research and potential applications in hardware accelerators.

Pros:

extensive experiments and well written

Cons:

Could use more explanation on some ablations